# Non-asymptotic approximations of Gaussian networks via second-order Poincaré inequalities

## Abstract

There is a growing interest on large-width asymptotic and non-asymptotic properties of deep Gaussian neural networks (NNs), namely NNs with weights initialized as Gaussian distributions. For a Gaussian NN of depth $L \geq 1$ and width $n \geq 1$, a well-established result is that, as $n \to +\infty$, the NN's output converges in distribution to a Gaussian process. Recently, quantitative versions of this CLT have been obtained by exploiting the recursive structure of the NN and its infinitely wide Gaussian limit, showing that the NN's output converges to its Gaussian limit at the rate $n^{-1/2}$, in the 2-Wasserstein distance, as well as in some convex distances. In this paper, we investigate the use of second-order Gaussian Poincaré inequalities to obtain quantitive CLTs for the NN's output, showing their pros and cons in such a new field of application. For shallow Gaussian NNs, i.e. $L = 1$, we show how second-order Poincaré inequalities provide a powerful tool, reducing the problem of establishing quantitative CLTs to the algebraic problem of computing the gradient and the Hessian of the NN's output, and lead to the rate of convergence $n^{-1/2}$ in the 1-Wasserstein distance. Instead, for deep Gaussian NNs, i.e. $L \geq 2$, the use of second-order Poincaré inequalities turns out to be more problematic. By relying on exact computations of the gradient and the Hessian of the NN's output, which is a non-trivial task due to its (algebraic) complexity that increases with $L$, we show that for $L = 2$ second-order Poincaré inequalities still lead to a quantitative CLT in the 1-Wasserstein distance, though with the rate of convergence $n^{-1/4}$, and we conjecture the same rate for any depth $L \geq 2$. Such a worsening in the rate is a peculiar feature of the use of second-order Poincaré inequalities, which are designed to be applied directly to the NN's output as a function of all the previous layers, hence not exploiting the recursive structure of the NN and/or its infinitely wide Gaussian limit. While this is a negative result over the state-of-the-art, it does not diminish the effectiveness of second-order Poincaré inequalities, which we prove to maintain their effectiveness in establishing a quantitative CLT for a complicated functional of Gaussian processes such as the deep Gaussian NN.

## 1 Introduction

To define a deep Gaussian neural networks (NN), let $\tau : \mathbb{R} \to \mathbb{R}$ be an activation function or nonlinearity, let $\mathbf{X}$ be a $d \times p$ input matrix, for any $d \geq 1$ and $p \geq 1$, with $\mathbf{x}_j$ being the $j$-th input row and $\boldsymbol{x}_u$ being the $u$-th input column, and for any $L \geq 1$ and $n \geq 1$ consider the following random variables: i) $(\mathbf{W}^{(0)}, \ldots, \mathbf{W}^{(L-1)}, \mathbf{w})$ independent random weights such that $\mathbf{W}^{(l)} = (w_{i,j_l}^{(l)})$ with the $w_{i,j_l}^{(l)}$ 's being i.i.d. as Gaussian $\mathcal{N}(0, \sigma_w^2)$ for $l = 0, \ldots, L-1$, $1 \leq i \leq n, 1 \leq j_0 \leq d, 1 \leq j_l \leq n$ for $l \geq 1$, and $\mathbf{w} = (w_1, \ldots, w_n)$ with the $w_i$'s being i.i.d as Gaussian $\mathcal{N}(0, \sigma_w^2)$ for $i = 1, \ldots, n$; ii) $(\mathbf{b}^{(0)}, \ldots, \mathbf{b}^{(L-1)}, b)$ independent random biases such that $\mathbf{b}^{(l)} = (b_1^{(l)}, \ldots, b_n^{(l)})$ with the $b_i^{(l)}$ 's being i.i.d. as $\mathcal{N}(0, \sigma_b^2)$ for $i = 1, \ldots, n$ and $l = 0, \ldots, L-1$, and with $b$ being $\mathcal{N}(0, \sigma_b^2)$. A (fully connected feed-forward) deep Gaussian NN of depth $L$ and width $n$ is the sequence $(f_i^{(l)}(\mathbf{X}, n))_{1 \leq i \leq n, 1 \leq l \leq L+1}$ defined recursively

as

$$
\begin{cases}
f_i^{(1)}(\mathbf{X}) = \sum_{j=1}^{d} w_{i,j}^{(0)} \mathbf{x}_j + b_i^{(0)} \mathbf{1}^T \\[2mm]
f_i^{(l)}(\mathbf{X}, n) = \frac{1}{\sqrt{n}} \sum_{j=1}^{n} w_{i,j}^{(l-1)} (\tau \circ f_j^{(l-1)}(\mathbf{X}, n)) + b_i^{(l-1)} \mathbf{1}^T \qquad l = 2, \ldots, L \\[2mm]
f^{(L+1)}(\mathbf{X}, n) = b + \frac{1}{\sqrt{n}} \sum_{i=1}^{n} w_i \tau(f_i^{(L)}(\mathbf{X}, n)),
\end{cases} \tag{1}
$$

where $\mathbf{1}$ is a $p$ dimensional column vector of 1's and $\circ$ denotes element-wise application. Let $(f_i^{(l)}(\mathbf{X}, n))_{i \geq 1}$ be the sequence obtained by extending $(\mathbf{W}^{(0)}, \ldots, \mathbf{W}^{(L-1)})$ and $(\mathbf{b}^{(0)}, \ldots, \mathbf{b}^{(L-1)})$ to infinite independent arrays, for $l = 0, \ldots, L-1$. Under the assumption that the activation function $\tau$ is continuous and satisfies $|\tau(s)| \leq \alpha + \beta|s|$ for every $s \in \mathbb{R}$ and some $\alpha, \beta \geq 0$, Matthews et al. (2018, Theorem 4) showed that as $n \to +\infty$ jointly over the first $l$ NN's layers

$$(f_i^{(l)}(\mathbf{X}, n))_{i \geq 1} \xrightarrow{w} (f_i^{(l)}(\mathbf{X}))_{i \geq 1},$$

where $(f_i^{(l)}(\mathbf{X}))_{i \geq 1}$, as a stochastic process indexed by $\mathbf{X}$, is distributed according to the product measure of $p$-dimensional Gaussian measures, namely $\otimes_{i \geq 1} \mathcal{N}_p(\mathbf{0}, \Sigma^{(l)})$, with the covariance matrix $\Sigma^{(l)}$ having the $(u, v)$-th entry defined recursively as follows: $\Sigma_{u,v}^{(1)} = \sigma_b^2 + \sigma_w^2 \langle \boldsymbol{x}_u, \boldsymbol{x}_v \rangle$ and $\Sigma_{u,v}^{(l)} = \sigma_b^2 + \sigma_w^2 \mathbb{E}[\tau(f)\tau(g)]$, where $(f, g)$ is a centered bivariate Gaussian vector with the covariance matrix given by $\Sigma_{u,u}^{(l-1)} e_1 e_1^T + \Sigma_{u,v}^{(l-1)}(e_1 e_2^T + e_2 e_1^T) + \Sigma_{v,u}^{(l-1)} e_2 e_2^T$, with $(e_1, e_2)$ being the canonical basis of $\mathbb{R}^2$. The infinitely wide Gaussian limit of the output $f^{(L+1)}(\mathbf{X}, n)$ follows as a corollary.

The work of Matthews et al. (2018) generalizes previous results of Neal (1996), characterizing the large-width asymptotic behaviour of a single-layer or shallow Gaussian NN, and of Lee et al. (2018b), which also investigate the large-width asymptotic behaviour of a deep Gaussian NN, though assuming a "sequential growth" of the width over the NN's layers. For generalizations and refinements of Matthews et al. (2018) we refer, among others, to the works of Garriga-Alonso et al. (2018); Lee et al. (2018a); Novak et al. (2018); Antognini (2019); Du et al. (2019); Yang (2019b); Aitken & Gur-Ari (2020); Andreassen & Dyer (2020); Bracale et al. (2021); Favaro et al. (2022); Lee et al. (2022); Hanin (2023); Mei & Montanari (2022). The large-width asymptotic behaviour of deep Gaussian NNs has been exploited in many directions: i) Bayesian inference for Gaussian processes arising from infinitely wide Gaussian NNs (Damianou & Lawrence, 2013; Garriga-Alonso et al., 2018; Lee et al., 2018a; Yaida, 2020; Hanin & Zlokapa, 2023); ii) kernel regression for infinitely wide Gaussian NNs trained with gradient descent through the neural tangent kernel (Hanin, 2018; Jacot et al., 2018; Du et al., 2019; Arora et al., 2019; Lee et al., 2019; Yang, 2019a; Yaida, 2020; Yang, 2020; Yang & Littwin, 2021; Roberts et al., 2022); iii) statistical analysis of infinitely wide Gaussian NNs as functions of the depth via information propagation (Poole et al., 2016; Schoenholz et al., 2017; Hayou et al., 2019; Hanin & Rolnick, 2018; Yaida, 2020; Roberts et al., 2022; Hanin, 2022). We also mention a line of research investigating properties of deep Gaussian NNs at finite width, which includes the works of Hanin & Nica (2019); Noci et al. (2021); Zavatone-Veth & Pehlevan (2021); Hanin & Zlokapa (2023); Roberts et al. (2022); Yaida (2020); Hanin (2018).

There has been a recent interest in establishing quantitative CLTs for the output of a deep Gaussian NN, with respect to suitable distances, providing the rate of convergence of $f^{(L+1)}(\mathbf{X}, n)$ to its infinitely wide limit. This problem was first investigated by Eldan et al. (2021) in the setting of shallow NNs, i.e. $L = 1$. They considered a shallow NN on the $(d-1)$-sphere with Gaussian distributed $w_{i,j}$'s and Rademacher distributed $w_i$'s, and assuming a polynomial activation function they established a functional quantitative CLT in the 2-Wasserstein distance $d_{W_2}$ for the NN's output. Still for shallow NNs on the $(d-1)$-sphere, the result of Eldan et al. (2021) has been generalized (and refined) in Klukowski (2022), assuming that the $w_{i,j}$'s are Uniformly distributed and the $w_i$'s have general distribution with finite fourth moment, and in Cammarota et al. (2023) assuming Gaussian weights. In the more general setting of Gaussian NNs, i.e. $L \geq 2$, Basteri & Trevisan (2022) first established a quantitative CLT in $d_{W_2}$ for the NN's output $f^{(L+1)}(\mathbf{X}, n)$. Their approach relies on: i) a preliminary estimate of the distance $d_{W_2}$ between $f^{(L+1)}(\mathbf{X}, n)$ and its infinitely wide Gaussian limit, through a triangular inequality that exploits the infinitely wide Gaussian limit of the NN; ii) the estimation of the terms in the inequality by an inductive approach that exploits the recursive structure of the NN in combination with properties of $d_{W_2}$. In particular, if $N \sim \mathcal{N}(\mathbf{0}, \Sigma^{(L+1)})$ is the infinitely wide limit of the NN's output $f^{(L+1)}(\mathbf{X}, n)$, then, assuming a Lipschitz activation function $\tau$, Basteri & Trevisan (2022, Theorem 1.1)

shows that

$$d_{W_2}(f^{(L+1)}(\mathbf{X}, n), N) \leq CL\sqrt{\frac{p}{n}}, \tag{2}$$

where $C \in (0, +\infty)$ is a constant depending only on $\tau$, $\mathbf{X}$ and $L$. Apollonio et al. (2023) and Favaro et al. (2023) generalized the result of Basteri & Trevisan (2022) to general convex distances and weaker hypotheses on $\tau$, providing upper bounds with the same rate of convergence $n^{-1/2}$, and (presumably) better constants. See also Balasubramanian et al. (2023) for further generalizations to NNs with non-Gaussian weights. As in Basteri & Trevisan (2022), the results of Apollonio et al. (2023) and Favaro et al. (2023) rely on triangular inequalities that exploit the recursive structure of the NN and its infinitely wide Gaussian limit, whose terms are estimated by different techniques, depending on the distance.

## 1.1 Our contributions

In this paper, we investigate the use of second-order Poincaré inequalities to establish quantitative CLTs for the NN's output $f^{(L+1)}(\mathbf{X}, n)$, showing their pros and cons in such a new field of application. Second-order Poincaré inequalities were first introduced in Chatterjee (2009) and Nourdin et al. (2009) as a general tool to obtain quantitative CLTs for functionals of Gaussian process, by estimating suitable distances or approximation errors between the functional of interest and a Gaussian process. This is precisely the setting of deep Gaussian NNs. In particular, we consider the use of some recent refinements of second-order Poincaré inequalities, which provide tight estimates of the approximation error, with (presumably) optimal rates (Vidotto, 2020). To introduce our results, it is useful to consider the setting of shallow NNs. In particular, let $d_{W_1}$ be the 1-Wasserstein distance and consider a shallow Gaussian NN with a 1-dimensional unitary input, i.e. $d = 1$ and $x = 1$, unit variance's weight, i.e. $\sigma_w^2 = 1$, and no biases, i.e. $b_i^{(0)} = b = 0$ for any $i \geq 1$. If $N \sim \mathcal{N}(0, \sigma^2)$ is the infinitely wide limit of the NN's output $f^{(2)}(1, n)$, then assuming the activation function $\tau \in C^2(\mathbb{R})$ such $\tau$ and its first and second derivatives are bounded above by $\alpha + \beta|x|^\gamma$, for $\alpha, \beta, \gamma \geq 0$, we show that

$$d_{W_1}(f^{(2)}(1, n), N) \leq C\frac{1}{\sqrt{n}}, \tag{3}$$

where $C$ is a constant depending only on $\tau$, which is computed explicitly. We show that a result analogous to (3) holds true for a shallow Gaussian NN with bias, with the approximation error being in the 1-Wasserstein distance, the total variation distance and the Kolmogorov-Smirnov distance, and then we generalize (3) to the setting of shallow Gaussian NNs with $p > 1$ inputs. For shallow NNs, our results show how second-order Poincaré inequalities provide a powerful tool to estimate the distance between the NN's output and its infinitely wide Gaussian limit. They reduce the problem to algebraic calculations for the gradient and the Hessian of the NN's output, which are straightforward for shallow NNs, leading to the same rate $n^{-1/2}$ as in Basteri & Trevisan (2022), Apollonio et al. (2023) and Favaro et al. (2023)

We also consider the use of second-order Poincaré inequalities to establish quantitative CLTs in the more general setting of deep Gaussian NNs. In principle, for $L \geq 2$ second-order Poincaré inequalities may be applied to the NN's output $f^{(L+1)}(\mathbf{X}, n)$ along the same lines as for $L = 1$. However, for $L \geq 2$ such a direct application of second-order Poincaré inequalities turns out to be more problematic, leading to a worst rate of convergence than $n^{-1/2}$, which is expected from Basteri & Trevisan (2022), Apollonio et al. (2023) and Favaro et al. (2023). As for shallow NNs, our results rely on the computation of the gradient and the Hessian of the NN's output, which is a non-trivial task for $L \geq 2$, due to an (algebraic) complexity that increases with the depth $L$. We make this computation exactly, and apply the resulting expressions of the gradient and the Hessian to obtain an (implicit) estimate of the distance $d_{W_1}$ between $f^{(L+1)}(\mathbf{X}, n)$ and its infinitely wide Gaussian limit. As an example, we obtain an explicit estimate for $L = 2$. In particular, if $N \sim \mathcal{N}(0, \Sigma^{(3)})$ is the infinitely wide limit of the NN's output $f^{(3)}(\mathbf{X}, n)$, then assuming the activation function $\tau \in C^2(\mathbb{R})$ such $\tau$ and its first and second derivatives are bounded above by $\alpha + \beta|x|^\gamma$, for $\alpha, \beta, \gamma \geq 0$, we show that

$$d_{W_1}(f^{(3)}(\mathbf{X}, n), N) \leq CL\sqrt{\frac{p}{\sqrt{n}}}, \tag{4}$$

where $C \in (0, +\infty)$ is a constant that depends only on $\tau$, $\mathbf{X}$ and $L$. In general, we conjecture that the same rate of convergence $n^{-1/4}$ holds true for any depth $L \geq 2$. Differently from Basteri & Trevisan (2022), Apollonio et al. (2023) and Favaro et al. (2023), our approach does not rely on the use of triangular inequalities that exploit the recursive structure of the NN and/or its infinitely wide Gaussian limit, since second-order Poincaré inequalities are designed to

be applied directly to $f^{(L+1)}(\mathbf{X}, n)$, as a function of all the NN's weights. This is arguably what determines the worsening in the rate of convergence with respect to (4). While this is a negative result over the state-of-the-art in the field, it does not diminish the effectiveness of second-order Poincaré inequalities, which we prove to maintain their effectiveness in establishing a quantitative CLT for a complicated functional of Gaussian processes such as the deep Gaussian NN.

## 1.2 Organization of the paper

The paper is structured as follows. In Section 2 we present a brief overview on second-order Poincaré inequalities, recalling some results of Vidotto (2020) that are critical to provide quantitative CLTs for deep Gaussian NNs. Section 3 contains our results in the setting of shallow Gaussian NNs, whereas in Section 4 we extend these results to the setting of deep Gaussian NNs. In Section 5 we present some numerical illustrations of our results for shallow NNs, and Section 6 we discuss our results and present some directions of future research. Proofs of our results are deferred to the Appendix.

## 2 Preliminaries on second-order Poincaré inequalities

Through the paper, we denote by $(\Omega, \mathcal{F}, \mathbb{P})$ the (standard) probability space on which random variables are assumed to be defined. Moreover, we make use of the notation $\|X\|_{L^q} := (\mathbb{E}[X^q])^{1/q}$ for the $L^q$ norm of the random variable $X$. In this work, we consider some popular distances between (probability) distributions of real-valued random variables. In particular, let $X$ and $Y$ be two random variables in $\mathbb{R}^d$, for some $d \geq 1$. We denote by $d_{W_1}$ the 1-Wasserstein distance, that is,

$$d_{W_1}(X, Y) = \sup_{h \in \mathscr{H}} |\mathbb{E}[h(X)] - \mathbb{E}[h(Y)]|,$$

where $\mathscr{H}$ is the class of all functions $h : \mathbb{R}^d \to \mathbb{R}$ such that it holds true that $\|h\|_{\mathrm{Lip}} \leq 1$, with $\|h\|_{\mathrm{Lip}} = \sup_{x,y \in \mathbb{R}^d, x \neq y} |h(x) - h(y)|/\|x - y\|_{\mathbb{R}^d}$. Furthermore, we denote by $d_{TV}$ the total variation distance, which is defined as

$$d_{TV}(X, Y) = \sup_{B \in \mathscr{B}(\mathbb{R}^m)} |\mathbb{P}(X \in B) - \mathbb{P}(Y \in B)|,$$

where $\mathscr{B}(\mathbb{R}^d)$ is the Borel $\sigma$-field of $\mathbb{R}^d$. Finally, we denote by $d_{KS}$ the Kolmogorov-Smirnov distance, which is defined as

$$d_{KS}(X, Y) = \sup_{z_1, \ldots, z_d \in \mathbb{R}} |\mathbb{P}\left(X \in \times_{i=1}^d (-\infty, z_i]\right) - \mathbb{P}\left(Y \in \times_{i=1}^d (-\infty, z_i]\right)|.$$

We recall the following (well-known) interplays between some of the above distances: i) $d_{KS}(\cdot, \cdot) \leq d_{TV}(\cdot, \cdot)$; ii) if $X$ is a real-valued random variable and $N \sim \mathcal{N}(0, 1)$ is the standard Gaussian random variable then $d_{KS}(X, N) \leq 2\sqrt{d_{W_1}(X, N)}$.

Second-order Poincaré inequalities provide a tool for Gaussian approximation of functionals of Gaussian fields (Chatterjee, 2009; Nourdin et al., 2009). See also Nourdin & Peccati (2012) and references therein for a detailed account on second-order Poincaré inequalities. For our work, it is useful to recall some results developed in Vidotto (2020), which provide improved versions of the second-order Poincaré inequality first introduced in Chatterjee (2009) for random variables and then extended in Nourdin et al. (2009) to general infinite-dimensional Gaussian fields. Let $N \sim \mathcal{N}(0, 1)$. Second-order Poincaré inequalities can be seen as an iteration of the so-called Gaussian Poincaré inequality, which states that

$$\mathrm{Var}[f(N)] \leq \mathbb{E}[f'(N)^2] \tag{5}$$

for every differentiable function $f : \mathbb{R} \to \mathbb{R}$, a result that was first discovered in the seminal work of Nash (1956), and then reproved by Chernoff (1981). The inequality (5) implies that if the $L^2$ norm of the random variable $f'(N)$ is small, then so are the fluctuations of the random variable $f(N)$. The first version of a second-order Poincaré inequality was obtained in Chatterjee (2009), where it is proved that one can iterate (5) in order to assess the total variation distance between the distribution of $f(N)$ and the distribution of a Gaussian random variable with matching mean and variance.

**Theorem 2.1** (Chatterjee (2009) - second-order Poincaré inequality). *Let $X \sim \mathcal{N}(0, I_{d \times d})$. Take any $f \in C^2(\mathbb{R}^d)$, and $\nabla f$ and $\nabla^2 f$ denote the gradient of $f$ and Hessian of $f$, respectively. Suppose that $f(X)$ has a finite fourth moment, and let $\mu = \mathbb{E}[f(X)]$ and $\sigma^2 = \mathrm{Var}[f(X)]$. Let $N \sim \mathcal{N}(\mu, \sigma^2)$ then*

$$d_{TV}(f(X), N) \leq \frac{2\sqrt{5}}{\sigma^2} \left\{ \mathbb{E}\left[\|\nabla f(X)\|_{\mathbb{R}^d}^4\right] \right\}^{1/4} \left\{ \mathbb{E}\left[\|\nabla^2 f(X)\|_{op}^4\right] \right\}^{1/4}, \tag{6}$$

*where $\|\cdot\|_{op}$ stands for the operator norm of the Hessian $\nabla^2 f(X)$ regarded as a random $d \times d$ matrix.*

Nourdin et al. (2009) pointed out that the Stein-type inequalities that lead to (6) are special instances of a more general class of inequalities, which can be obtained by combining Stein's method and Malliavin calculus on an infinite-dimensional Gaussian space. In particular, Nourdin et al. (2009) obtained a more general version of (6), involving functionals of arbitrary infinite-dimensional Gaussian fields. Both (6) and its generalization in Nourdin et al. (2009) are known to give suboptimal rates of convergence. This is because, in general, it is not possible to obtain an explicit computation of the expectation of the operator norm involved in the estimate of total variation distance, which leads to move further away from the distance in distribution and use bounds on the operator norm instead of computing it directly. To overcome this drawback, the work of Vidotto (2020) adapted to the Gaussian setting an approach recently developed in Last et al. (2016) to obtain second-order Poincaré inequalities for Gaussian approximation of Poisson functionals, yielding estimates of the approximation error that are (presumably) optimal. The next theorem states Vidotto (2020, Theorem 2.1) for the special case of a function $f(X)$, with $f \in C^2(\mathbb{R}^d)$ such that its partial derivatives have sub-exponential growth, and $X \sim \mathcal{N}(0, I_{d \times d})$. See Appendix A for an overview of Vidotto (2020, Theorem 2.1).

**Theorem 2.2** (Vidotto (2020) - 1-dimensional second-order Poincaré inequality). *Let $F = f(X)$, for some $f \in C^2(\mathbb{R}^d)$, and $X \sim \mathcal{N}(0, I_{d \times d})$ such that $E[F] = 0$ and $E[F^2] = \sigma^2$. Let $N \sim \mathcal{N}(0, \sigma^2)$, then*

$$d_M(F, N) \leq c_M \sqrt{\sum_{l,m=1}^d \left\{ \mathbb{E}\left[\left(\langle \nabla_{l,\cdot}^2 F, \nabla_{m,\cdot}^2 F \rangle\right)^2\right] \right\}^{1/2} \left\{ \mathbb{E}\left[(\nabla_l F \nabla_m F)^2\right] \right\}^{1/2}}, \tag{7}$$

*where $\langle \cdot, \cdot \rangle$ is the scalar product, $M \in \{TV, KS, W_1\}$, $c_{TV} = \frac{4}{\sigma^2}$, $c_{KS} = \frac{2}{\sigma^2}$, $c_{W_1} = \sqrt{\frac{8}{\sigma^2 \pi}}$ and $\nabla_{i,\cdot}^2 F$ is the $i$-th row of the Hessian matrix of $F = f(X)$ while $\nabla_i F$ is the $i$-th element of the gradient of $F$.*

The next theorem generalizes Theorem 2.2 to multidimensional functionals. For $p > 1$, the next theorem states Vidotto (2020, Theorem 2.3) for the special case of a function $(f_1(X), \ldots, f_p(X))$, with $f_1, \ldots, f_p \in C^2(\mathbb{R}^d)$ such that its partial derivatives have sub-exponential growth, and $X \sim \mathcal{N}(0, I_{d \times d})$. See Appendix A for an overview of Vidotto (2020, Theorem 2.3).

**Theorem 2.3** (Vidotto (2020) - $p$-dimensional second-order Poincaré inequality). *For any $p > 1$ let $[F_1 \ \ldots \ F_p] = [f_1(X) \ \ldots \ f_p(X)]$, for some $f_1, \ldots, f_p \in C^2(\mathbb{R}^d)$, and $X \sim \mathcal{N}(0, I_{d \times d})$ such that $E[F_i] = 0$ for $i = 1, \ldots, p$ and $E[F_i F_j] = c_{ij}$ for $i, j = 1, \ldots, p$, with $C = \{c_{ij}\}_{i,j=1,\ldots,p}$ being a symmetric and positive definite matrix, i.e. a variance-covariance matrix. Let $N \sim \mathcal{N}(0, C)$, then*

$$d_{W_1}(F, N) \leq 2\sqrt{p} \, \|C^{-1}\|_2 \, \|C\|_2 \sqrt{\sum_{i,k=1}^p \sum_{l,m=1}^d \left\{ \mathbb{E}\left[\left(\langle \nabla_{l,\cdot}^2 F_i, \nabla_{m,\cdot}^2 F_i \rangle\right)^2\right] \right\}^{1/2} \left\{ \mathbb{E}\left[(\nabla_l F_k \nabla_m F_k)^2\right] \right\}^{1/2}} \tag{8}$$

*where $\|\cdot\|_2$ is the spectral norm of a matrix.*

## 3 Results for shallow Gaussian NNs

We make use of the second-order Poincaré inequalities of Section 2 to obtain quantitative CLTs for the NN's output $F := f^{L+1}(X, n)$, with $f^{L+1}(X, n)$ defined in (1), with $L = 1$. In particular, we provide a quantification of the approximation error between $F$ and its Gaussian limit, with respect to the 1-Wasserstein distance, the total variation distance and the Kolmogorov-Smirnov distance. We start with a NN with a 1-dimensional unitary input, i.e. $d = 1$

and $x = 1$, unit variance's weight, i.e. $\sigma_w^2 = 1$, and no biases, i.e. $b_i^{(0)} = b = 0$ for any $i \geq 1$. That is, we consider the NN

$$F := \frac{1}{n^{1/2}} \sum_{j=1}^{n} w_j \tau(w_j^{(0)}). \tag{9}$$

By means of a straightforward calculation, it follows that $\mathbb{E}[F] = 0$ and $\mathrm{Var}[F] = \mathbb{E}_{Z \sim \mathcal{N}(0,1)}[\tau^2(Z)]$. As the NN $F$ defined in (9) is a function of independent standard Gaussian random variables, Theorem 2.2 can be applied to approximate $F$ with a Gaussian random variable with the same mean and variance as $F$, quantifying the approximation error.

**Theorem 3.1.** *Let $F$ be the NN (9) with $\tau \in C^2(\mathbb{R})$ such that $|\tau(x)| \leq \alpha + \beta|x|^\gamma$ and $\left|\frac{d^l}{dx^l}\tau(x)\right| \leq \alpha + \beta|x|^\gamma$ for $l = 1, 2$ and some $\alpha, \beta, \gamma \geq 0$. If $N \sim \mathcal{N}(0, \sigma^2)$ with $\sigma^2 = \mathbb{E}_{Z \sim \mathcal{N}(0,1)}[\tau^2(Z)]$, then for any $n \geq 1$*

$$d_M(F, N) \leq \frac{c_M}{\sqrt{n}} \sqrt{3(1 + \sqrt{2})} \cdot \|\alpha + \beta|Z|^\gamma\|_{L^4}^2, \tag{10}$$

*where $Z \sim \mathcal{N}(0,1)$, $M \in \{TV, KS, W_1\}$, with corresponding constants $c_{TV} = 4/\sigma^2$, $c_{KS} = 2/\sigma^2$, and $c_{W_1} = \sqrt{8/\sigma^2\pi}$.*

See Appendix B for the proof of Theorem 3.1. The proof follows by a direct application of Theorem 2.2, reducing the problem to (algebraic) calculations for the gradient and the Hessian of the NN, which are straightforward in the setting of shallow NNs. In particular, the estimate (10) has the expected convergence rate $n^{-1/2}$ with respect to the 1-Wasserstein distance, the total variation distance and the Kolmogorov-Smirnov distance. As for the constant appearing in (10), it depends on the variance $\mathbb{E}_{Z \sim \mathcal{N}(0,1)}[\tau^2(Z)]$ of $F$. Once the activation function $\tau$ is specified, $\mathbb{E}_{Z \sim \mathcal{N}(0,1)}[\tau^2(Z)]$ can be evaluated by an exact or approximate computation, as well as by providing suitable lower bounds for it.

Now, we present an extension of Theorem 3.1 to a Gaussian NN with one input $x$, showing that the problem still reduces to a direct application of Theorem 2.2. In particular, it is convenient to write the output of the NN in (1) as follows:

$$F := \frac{1}{n^{1/2}} \sigma_w \sum_{j=1}^{n} w_j \tau(\sigma_w \langle w_j^{(0)}, x \rangle + \sigma_b b_j^{(0)}) + \sigma_b b, \tag{11}$$

with $w_j^{(0)} = [w_{j,1}^{(0)}, \ldots, w_{j,d}^{(0)}]^T$ and $w_j \overset{d}{=} w_{j,i}^{(0)} \overset{\text{iid}}{\sim} \mathcal{N}(0,1)$. In particular, we set $\Gamma^2 = \sigma_w^2 \|x\|^2 + \sigma_b^2$, and for $n \geq 1$ we consider a collection $(Y_1, \ldots, Y_n)$ of independent standard Gaussian random variables. Then, from (11) we can write

$$F \overset{d}{=} \frac{1}{n^{1/2}} \sigma_w \sum_{j=1}^{n} w_j \tau(\Gamma Y_j) + \sigma_b b.$$

As before, some straightforward calculations leads to $\mathbb{E}[F] = 0$ and $\mathrm{Var}[F] = \sigma_w^2 \mathbb{E}_{Z \sim \mathcal{N}(0,1)}[\tau^2(\Gamma Z)] + \sigma_b^2$. Since $F$ in (11) is a function of independent standard Gaussian random variables, Theorem 2.2 can be directly applied to approximate $F$ with a Gaussian random variable with the same mean and variance as $F$, quantifying the approximation error.

**Theorem 3.2.** *Let $F$ be the output of the NN (11) with $\tau \in C^2(\mathbb{R})$ such that $|\tau(x)| \leq \alpha + \beta|x|^\gamma$ and $\left|\frac{d^l}{dx^l}\tau(x)\right| \leq \alpha + \beta|x|^\gamma$ for $l = 1, 2$ and some $\alpha, \beta, \gamma \geq 0$. If $N \sim \mathcal{N}(0, \sigma^2)$ with $\sigma^2 = \sigma_w^2 \mathbb{E}_{Z \sim \mathcal{N}(0,1)}[\tau^2(\Gamma Z)] + \sigma_b^2$ and $\Gamma = (\sigma_w^2 \|x\|^2 + \sigma_b^2)^{1/2}$, then for any $n \geq 1$*

$$d_M(F, N) \leq \frac{c_M \sqrt{\Gamma^2 + \Gamma^4(2 + \sqrt{3(1 + 2\Gamma^2 + 3\Gamma^4)})} \|\alpha + \beta|\Gamma Z|^\gamma\|_{L^4}^2}{\sqrt{n}}, \tag{12}$$

*where $Z \sim \mathcal{N}(0,1)$, $M \in \{TV, KS, W_1\}$, with corresponding constants $c_{TV} = 4/\sigma^2$, $c_{KS} = 2/\sigma^2$, $c_{W_1} = \sqrt{8/\sigma^2\pi}$.*

See Appendix C for the proof of Theorem 3.2. As for Theorem 3.1, the estimate (12) of the approximation error $d_M(F, N)$ has the expected convergence rate $n^{-1/2}$ with respect to the 1-Wasserstein distance, the total variation distance and the Kolmogorov-Smirnov distance, with the constant depending on the variance $\sigma_w^2 \mathbb{E}_{Z \sim \mathcal{N}(0,1)}\left[\tau^2(\Gamma Z)\right] + \sigma_b^2$ of $F$. Within the setting of Theorem 3.2, the same rate of convergence $n^{-1/2}$ is obtained through different techniques in Basteri & Trevisan (2022, Theorem 1.1), Apollonio et al. (2023, Theorem 4.1) and Favaro et al. (2023, Theorem 3.3), possibly leading to different constants. Because of the definition of the NN $F$, analogous result follow from the classical Berry-Eseen theorem in the Kolmogorov-Smirnov distance, as well as the CLT in the Wasserstein distance (Rio, 2009).

We conclude our analysis of shallow NNs, by presenting an extension of Theorem 3.2 to a Gaussian NN output with $p > 1$ inputs $(\boldsymbol{x}_1, \ldots, \boldsymbol{x}_p)$, where $\boldsymbol{x_i} \in \mathbb{R}^d$ for $i = 1, \ldots, p$. In particular, we consider $F := [F_1 \ \ldots \ F_p]$, where

$$F_i := \frac{1}{n^{1/2}} \sigma_w \sum_{j=1}^n w_j \tau(\sigma_w \langle w_j^{(0)}, \boldsymbol{x_i} \rangle + \sigma_b b_j^{(0)}) + \sigma_b b, \tag{13}$$

with $w_j^{(0)} = [w_{j,1}^{(0)}, \ldots, w_{j,d}^{(0)}]^T$ and $w_j \stackrel{d}{=} w_{j,i}^{(0)} \stackrel{d}{=} b_j^{(0)} \stackrel{d}{=} b \stackrel{\text{iid}}{\sim} \mathcal{N}(0, 1)$. Under this setting of multivariate Gaussian distributions, Theorem 2.3 can be applied to obtain an approximation of $F$ with a Gaussian random vector whose mean and covariance are the same as $F$. The resulting estimate of the approximation error depends on the maximum and the minimum eigenvalues, i.e. $\lambda_1(C)$ and $\lambda_p(C)$ respectively, of the covariance matrix $C$, whose $(i, k)$-th entry is given by

$$\mathbb{E}[F_i F_k] = \sigma_w^2 \mathbb{E}[\tau(Y_i)\tau(Y_k)] + \sigma_b^2, \tag{14}$$

where $Y \sim \mathcal{N}(0, \sigma_w^2 \mathbf{X}^T \mathbf{X} + \sigma_b^2 \mathbf{1}\mathbf{1}^T)$, with $\mathbf{1}$ being the all-one vector of dimension $p$ and $\mathbf{X}$ being the $n \times p$ matrix of the inputs $\{\boldsymbol{x_i}\}_{i \in [p]}$.

**Theorem 3.3.** *Let $F = [F_1 \ \ldots \ F_p]$ with $F_i$ being the NN output in (13), for $i = 1, \ldots, p$, with $\tau \in C^2(\mathbb{R})$ such that $|\tau(x)| \leq \alpha + \beta|x|^\gamma$ and $\left|\frac{d^l}{dx^l}\tau(x)\right| \leq \alpha + \beta|x|^\gamma$ for $l = 1, 2$ and some $\alpha, \beta, \gamma \geq 0$. Furthermore, let $C$ be the covariance matrix of $F$, whose entries are given in (14), and define $\Gamma_i^2 = \sigma_w^2\|\boldsymbol{x_i}\|^2 + \sigma_b^2$ and $\Gamma_{ik} = \sigma_w^2 \sum_{j=1}^d |x_{ij} x_{kj}| + \sigma_b^2$. If $N = [N_1 \ \cdots \ N_p] \sim \mathcal{N}(0, C)$, then for any $n \geq 1$*

$$d_{W_1}(F, N) \leq 2\sigma_w^2 \tilde{K} \frac{\lambda_1(C)}{\lambda_p(C)} \sqrt{\frac{p}{n}}, \tag{15}$$

*where $\lambda_1(C)$ and $\lambda_p(C)$ are the maximum and the minimum eigenvalues of $C$, respectively, and where*

$$\tilde{K} = \left\{ \sum_{i,k=1}^p (\Gamma_i^2 + \sqrt{3(1 + 2\Gamma_i^2 + 3\Gamma_i^4)}\Gamma_{ik}^2 + 2\Gamma_i^2\Gamma_{ik})\|\alpha + \beta|\Gamma_i Z|^\gamma\|_{L^4}^2 \|\alpha + \beta|\Gamma_k Z|^\gamma\|_{L^4}^2 \right\}^{1/2},$$

*with $Z \sim \mathcal{N}(0, 1)$.*

See Appendix D for the proof of Theorem 3.3. Along the same lines of the proofs of Theorem 3.1 and Theorem 3.2, Theorem 3.3 follows by a direct application of Theorem 2.3, which boils down to straightforward (algebraic) calculations for the gradient and the Hessian of the NN. The estimate (19) of the approximation error $d_{W_1}(F, N)$ has the expected convergence rate $n^{-1/2}$ with respect to the 1-Wasserstein distance, with a constant depending on the spectral norms of the covariance matrix $C$ and the precision matrix $C^{-1}$. In particular, such spectral norms must be computed explicitly for the specific activation $\tau$ in use, or at least bounded from above, in order to apply Theorem 3.3. This boils down to finding the greatest eigenvalue $\lambda_1$ and the smallest eigenvalue $\lambda_p$ of the matrix $C$, which can be done for a broad class of activations with classical optimization techniques, or at least bounding $\lambda_1$ from above and $\lambda_p$ from below (Diaconis & Stroock, 1991; Guattery et al., 1999). Within the setting of Theorem 3.3, the same rate of convergence $n^{-1/2}$ is obtained through different techniques in Basteri & Trevisan (2022, Theorem 1.1), Apollonio et al. (2023, Theorem 6.1 and Theorem 6.2) and Favaro et al. (2023, Theorem 3.5), possibly leading to different constants.

## 4  Results for deep Gaussian NNs

Now, we consider the use of the second-order Poincaré inequalities of Section 2 to obtain quantitative CLTs for the output of a deep Gaussian NN, thus generalizing Theorem 3.2 and Theorem 3.3. In principle, for $L \geq 2$ second-order

Poincaré inequalities may be applied to the NN's output $f^{(L+1)}(\mathbf{X}, n)$ along the same lines as for $L = 1$, though at the cost of more involved algebraic calculations. However, we show that for $L \geq 2$ such a direct application of second-order Poincaré inequalities is more problematic, leading to a worst rate of convergence than $n^{-1/2}$, which is expected from Basteri & Trevisan (2022), Apollonio et al. (2023) and Favaro et al. (2023). As for shallow NNs, the use of second-order Poincaré inequalities rely on the computation of the gradient and the Hessian of the NN's output, which for $L \geq 2$ is a non-trivial task due to its (algebraic) complexity that increases with the depth $L$. In particular, let $F := f^{(L+1)}(\mathbf{X}, n)$, with $f^{(L+1)}(\mathbf{X}, n)$ as in (1). Since $F$ is a function of i.i.d. Gaussian random weights, then Theorem 2.3 can be applied to give an upper bound for the 1-Wasserstein distance between $F$ and a Gaussian random vector with the same covariance matrix. See Appendix E for explicit expressions of the gradient and the Hessian of the NN output.

**Theorem 4.1** (Multi-layer NN bound). *Let $F = [F_1 \ \ldots \ F_p] := f^{(L+1)}(\mathbf{X}, n)$ with $f^{(L+1)}(\mathbf{X}, n)$ being the output of the NN defined in (1), and let $C$ be the covariance matrix of $F$. If $N = [N_1 \ \cdots \ N_p] \sim \mathcal{N}(0, C)$, then for any $n \geq 1$ and $L > 1$*

$$
d_{W_1}(F, N) \leq 2\sqrt{p} \frac{\lambda_1(C)}{\lambda_p(C)}
$$

$$
\times \left\{ \sum_{i,k=1}^{p} \sum_{l,m=0}^{L-1} \sum_{i_1, i_2, i_3, i_4 = 1}^{n} \left\{ \mathbb{E}\left[ \left( \left\langle \nabla^2_{w^{(l)}_{i_1, i_2}, \cdot} F_i, \nabla^2_{w^{(m)}_{i_3, i_4}, \cdot} F_k \right\rangle \right)^2 \right] \mathbb{E}\left[ \left( \frac{\partial F_i}{\partial w^{(l)}_{i_1, i_2}} \frac{\partial F_k}{\partial w^{(m)}_{i_3, i_4}} \right)^2 \right] \right\}^{1/2} \right.
$$

$$
+ 2 \sum_{i,k=1}^{p} \sum_{l=0}^{L-1} \sum_{i_1, i_3, i_4 = 1}^{n} \left\{ \mathbb{E}\left[ \left( \left\langle \nabla^2_{w_{i_1}, \cdot} F_i, \nabla^2_{w^{(l)}_{i_3, i_4}, \cdot} F_k \right\rangle \right)^2 \right] \mathbb{E}\left[ \left( \frac{\partial F_i}{\partial w_{i_1}} \frac{\partial F_k}{\partial w^{(l)}_{i_3, i_4}} \right)^2 \right] \right\}^{1/2}
$$

$$
\left. + \sum_{i,k=1}^{p} \sum_{i_1, i_3 = 1}^{n} \left\{ \mathbb{E}\left[ \left( \left\langle \nabla^2_{w_{i_1}, \cdot} F_i, \nabla^2_{w_{i_3}, \cdot} F_k \right\rangle \right)^2 \right] \mathbb{E}\left[ \left( \frac{\partial F_i}{\partial w_{i_1}} \frac{\partial F_k}{\partial w_{i_3}} \right)^2 \right] \right\}^{1/2} \right\}^{1/2} .
$$

The estimate of $d_{W_1}(F, N)$ in Theorem 4.1 is implicit in nature, because controlling the expectations involving the gradient and the Hessian of the NN is a non-trivial task for a general depth $L \geq 2$. This is a computational issue arising from the use of second-order Poincaré inequalities for $L \geq 2$. For example, in the case $p = 1$, with $\mathbf{X} = \boldsymbol{x}$,

$$
\mathbb{E}\left[ \left( \frac{\partial F}{\partial w_i} \frac{\partial F}{\partial w_j} \right)^2 \right] = \left( \frac{\sigma_w}{\sqrt{n}} \right)^4 \mathbb{E}\left[ \tau\left( f^{(L)}_i(\boldsymbol{x}, n) \right)^2 \tau\left( f^{(L)}_j(\boldsymbol{x}, n) \right)^2 \right]. \tag{16}
$$

As the random variables on the right-hand side of (16) are dependent, to deal with the expectation one may consider to condition with respect to the output of the previous layer, in this case the vector $f^{(L-1)}_\cdot(\boldsymbol{x}, n) := (f^{(L-1)}_j(\boldsymbol{x}, n) : j \in [n])$, and then use the fact that $f^{(L)}_i(\boldsymbol{x}, n)$ and $f^{(L)}_j(\boldsymbol{x}, n)$ are conditionally i.i.d. given $f^{(L-1)}_\cdot(\boldsymbol{x}, n)$. Then, (16) factorizes as

$$
\mathbb{E}\left[ \left( \frac{\partial F}{\partial w_i} \frac{\partial F}{\partial w_j} \right)^2 \right]
$$

$$
= \left( \frac{\sigma_w}{\sqrt{n}} \right)^4 \mathbb{E}\left[ \tau\left( f^{(L)}_i(\boldsymbol{x}, n) \right)^2 \tau\left( f^{(L)}_j(\boldsymbol{x}, n) \right)^2 \right]
$$

$$
= \left( \frac{\sigma_w}{\sqrt{n}} \right)^4 \mathbb{E}\left[ \mathbb{E}\left[ \tau\left( f^{(L)}_i(\boldsymbol{x}, n) \right)^2 \tau\left( f^{(L)}_j(\boldsymbol{x}, n) \right)^2 \, \middle| \, f^{(L-1)}_\cdot(\boldsymbol{x}, n) \right] \right]
$$

$$
\overset{\text{cond. i.i.d.}}{=} \left( \frac{\sigma_w}{\sqrt{n}} \right)^4 \mathbb{E}\left[ \mathbb{E}\left[ \tau\left( f^{(L)}_i(\boldsymbol{x}, n) \right)^2 \, \middle| \, f^{(L-1)}_\cdot(\boldsymbol{x}, n) \right]^2 \right],
$$

which, however, is not helpful, since the distribution of the random variable $f^{(L-1)}(\boldsymbol{x}, n)$ is not Gaussian. The only exception is in the case of a NN with two hidden layers, i.e. $L = 2$, where the conditioning argument provides an effective way to bound the expectations, being the random variable $f^{(L-1)}(\boldsymbol{x}, n) = f^{(1)}(\boldsymbol{x})$ distributed as a Gaussian distribution.

We conclude by applying the conditioning argument to make more explicit the estimate of $d_{W_1}(F, N)$ of Theorem 4.1 for $L = 2$. For simplicity, we assume a NN without bias. Given an input $\boldsymbol{x} \in \mathbb{R}^d$, the output $F$ takes the functional form

$$F = \sigma_w n^{-1/2} \sum_{i=1}^n w_i \tau \left( \sigma_w n^{-1/2} \sum_{j=1}^n w_{i,j}^{(1)} \tau(\sigma_w \langle w_j^{(0)}, \boldsymbol{x} \rangle_{\mathbb{R}^d}) \right). \tag{17}$$

As before, $F \stackrel{d}{=} \tilde{F}$, where

$$\tilde{F} := \sigma_w n^{-1/2} \sum_{i=1}^n w_i \tau \left( \sigma_w n^{-1/2} \sum_{j=1}^n w_{i,j}^{(1)} \tau(\Gamma Y_j) \right),$$

with $\Gamma^2 = \sigma_w^2 \|\boldsymbol{x}\|_2^2$ and $Y_j \stackrel{d}{=} w_{j,i}^{(1)} \stackrel{d}{=} w_j \sim \mathcal{N}(0, 1)$ for all $i, j \in [n]$. The next theorem applies Theorem 4.1 to establish the rate of convergence for the output of the Gaussian NN with one input of dimension $d$ and two hidden layers.

**Theorem 4.2** (2-hidden-layers NN bound). *Let $F$ be the NN output (17) with $\tau \in C^2(\mathbb{R})$ such that $|\tau(x)| \leq \alpha + \beta|x|^\gamma$ and $\left|\frac{d^l}{dx^l}\tau(x)\right| \leq \alpha + \beta|x|^\gamma$ for $l = 1, 2$ and some $\alpha, \beta, \gamma \geq 0$. If $N \sim \mathcal{N}(0, \sigma^2)$ with $\sigma^2 = \mathrm{Var}[F]$, then for any $n \geq 1$*

$$d_M(F, N) \leq c_M \frac{K_1}{\sqrt[4]{n}}, \tag{18}$$

*where $K_1$ is a constant independent of $n$ and $d$ which depends on some expectations of the standard Gaussian law and can be computed explicitly, and $c_M$ is as in Theorem 3.1.*

Theorem 4.2 can be adapted to a NN with $p$ inputs, in analogy to Theorem 3.3. The next theorem applies Theorem 4.1 to establish the rate of convergence for the output of the Gaussian NN with $p$ inputs of dimension $d$ and two hidden layers.

**Theorem 4.3.** *Let $F = [F_1 \ \ldots \ F_p]$ with $F_i$ being the NN output (17), for $i = 1, \ldots, p$, with $\tau \in C^2(\mathbb{R})$ such that $|\tau(x)| \leq \alpha + \beta|x|^\gamma$ and $\left|\frac{d^l}{dx^l}\tau(x)\right| \leq \alpha + \beta|x|^\gamma$ for $l = 1, 2$ and some $\alpha, \beta, \gamma \geq 0$. Furthermore, let $C$ be the covariance matrix of $F$. If $N = [N_1 \ \ldots \ N_p] \sim \mathcal{N}(0, C)$, then for any $n \geq 1$*

$$d_{W_1}(F, N) \leq 2K_p \frac{\lambda_1(C)}{\lambda_p(C)} \sqrt{\frac{p}{\sqrt{n}}} \tag{19}$$

*where $\lambda_1(C)$ and $\lambda_p(C)$ are the maximum and the minimum eigenvalues of $C$, respectively, and where $K_p$ is a constant independent of $n$ and $d$ which depends on some expectations of the standard Gaussian law and can be computed explicitly.*

See Appendix F for the proof of Theorem 4.2 and Theorem 4.3. These results show how the use of second-order Poincaré inequalities leads to a worse rate of convergence than the rate $n^{-1/2}$ established in Basteri & Trevisan (2022, Theorem 1.1), Apollonio et al. (2023, Theorem 6.1 and Theorem 6.2) and Favaro et al. (2023, Theorem 3.3. and Theorem 3.5). Differently from the works of Basteri & Trevisan (2022), Apollonio et al. (2023) and Favaro et al. (2023), our approach does not rely on the use of triangular inequalities that exploit the recursive structure of the NN and/or its infinitely wide Gaussian limit, since second-order Poincaré inequalities are designed to be applied directly to $f^{(L+1)}(\mathbf{X}, n)$, as a function of all the NN's layers. This is arguably what determines the worsening in the rate of convergence. For linearly-bounded activation functions, the direct use of second-order Gaussian Poincaré leads to the rate

$$\mathcal{O}\left(\sqrt{\frac{p}{\sqrt{n}}}\right),$$

and such a rate can not be improved, since assuming $\tau = id$ leads to the same rate. See Appendix F for details. Based on these observations, for a deep Gaussian NN of depth $L \geq 1$, we conjecture that Theorem 4.1 leads to the rate of convergence

$$\mathcal{O}\left(L\sqrt{\frac{p}{\sqrt{n}}}\right),$$

which is worse than the rate of convergence established, for instance, in Basteri & Trevisan (2022, Theorem 1.1), that is

$$\mathcal{O}\left(L\sqrt{\frac{p}{n}}\right).$$

As we proved for the activation function $\tau = id$, there are no chances to avoid this worsening in the rate of convergence when second order Poincaré inequalities are applied directly to the NN's output in order to establish a quantitative CLT for it.

## 5 Numerical illustrations

We present a simulation study with respect to two choices of the activation function $\tau$: i) $\tau(x) = \tanh x$, which is polynomially bounded with parameters $\alpha = 1$ and $\beta = 0$; ii) $\tau(x) = x^3$, which is polynomially bounded with parameters $\alpha = 6$, $\beta = 1$ and $\gamma = 3$. Each of the plots below is obtained as follows: for a fixed width of $n = k^3$, with $k \in \{1, \cdots, 16\}$, we simulate 5000 points from a single-layer NN as in Theorem 3.1 to produce an estimate of the distance between the NN and a Gaussian random variable with mean $0$ and variance $\sigma^2$, which is estimated by means of a Monte-Carlo approach. Estimates of the KS and TV distance are produced by means of the functions *KolmogorovDist* and *TotVarDist* from the package **distrEx** by Ruckdeschel et al. (2006) while those of the 1-Wasserstein distance using the function *wasserstein1d* from the package **transport** by Schuhmacher et al. (2022). We repeat this procedure 2000 times for every fixed $n \in \{3, 6, \cdots, 51\}$, compute the sample mean (blue dots), and compare these estimates with the theoretical explicit bound given by Theorem 3.1 (green dots), and with the implicit bound given by Theorem 4.1 (red dots).

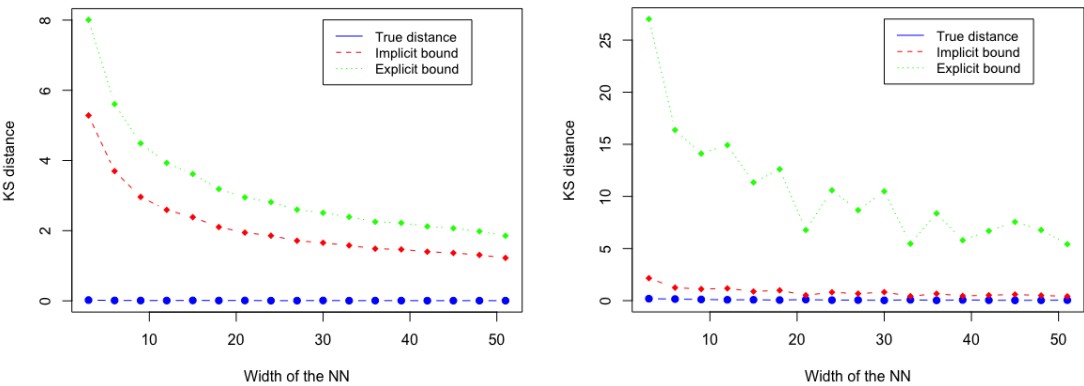

Figure 1: Estimates of the Kolmogorov-Smirnov distance for a Shallow NN of varying width $n \in \{3, 6, \cdots, 51\}$, with $\tau(x) = \tanh x$ (left) and $\tau(x) = x^3$ (right).

All the figures confirm that the distance between a shallow NN and an arbitrary Gaussian random variable, with the same mean and variance, is $\lesssim n^{-1/2}$, with approximation errors improving as the width $n \to \infty$. The evaluation of the implicit bound of Theorem 4.1 results in much tighter estimate of the distance than what provided by the explicit bound, which highlight the rate $n^{-1/2}$ at the cost of having a looser constant. This is clear in the case $\tau(x) = x^3$, where the polynomial envelope assumption leads to a much rougher bound to the one you may get computing the derivatives explicitly.

## 6 Discussion

We applied second-order Poincaré inequalities to establish quantitative CLTs for the NN's output $f^{(L+1)}(\mathbf{X}, n)$, showing their pros and cons in such a new field of application. For shallow Gaussian NNs, i.e. $L = 1$, Theorem 3.1, Theorem 3.2 and Theorem 3.3 show how second-order Poincaré inequalities provide a powerful tool: they reduce the problem of establishing quantitive CLTs to the algebraic problem of computing the gradient and the Hessian of the NN's output, which is straightforward for shallow NNs, and they lead to the rate of convergence $n^{-1/2}$ in the

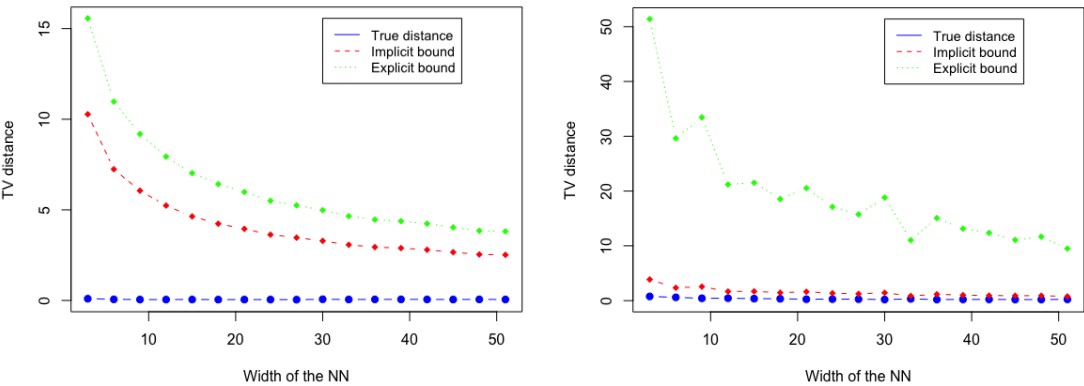

Figure 2: Estimates of the Total Variation distance for a Shallow NN of varying width $n \in \{3, 6, \cdots, 51\}$, with $\tau(x) = \tanh x$ (left) and $\tau(x) = x^3$ (right).

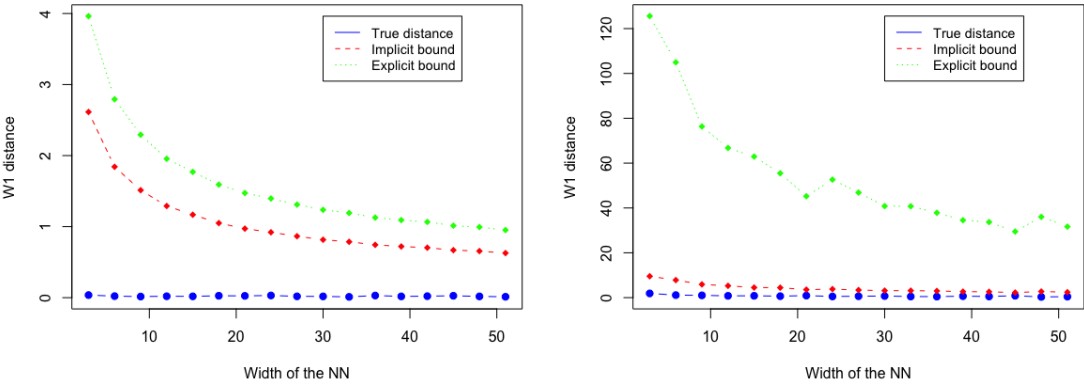

Figure 3: Estimates of the 1-Wasserstein distance for a Shallow NN of varying width $n \in \{3, 6, \cdots, 51\}$, with $\tau(x) = \tanh x$ (left) and $\tau(x) = x^3$ (right).

1-Wasserstein distance. This is the same rate of convergence obtained in Basteri & Trevisan (2022), Apollonio et al. (2023) and Favaro et al. (2023) by means of different techniques. Instead, for deep Gaussian NNs, i.e. $L \geq 2$, the use of second-order Poincaré inequalities is more problematic, leading to a worse rate of convergence than the rate $n^{-1/2}$ obtained in Basteri & Trevisan (2022), Apollonio et al. (2023) and Favaro et al. (2023). By relying on exact computations of the gradient and the Hessian of the NN's output, which is a non-trivial task due to its (algebraic) complexity that increases with $L$, Theorem 4.2 and Theorem 4.3 show that for $L = 2$ second-order Poincaré inequalities still lead to a quantitive CLT in the 1-Wasserstein distance, though with the rate of convergence $n^{-1/4}$. Differently from the works of Basteri & Trevisan (2022), Apollonio et al. (2023) and Favaro et al. (2023), our approach does not rely on the use of triangular inequalities that exploit the recursive structure of the NN and/or its infinitely wide Gaussian limit, since second-order Poincaré inequalities are designed to be applied directly to $f^{(L+1)}(\mathbf{X}, n)$, as a function of all the NN's weights.

Related to the choice of the activation function $\tau$, one may consider the problem of relaxing the hypothesis of polynomially boundedness and use a whatever $\tau \in C^2(\mathbb{R})$. Theorem 2.2 and Theorem 2.3 would still apply, with the only difference that the bound would be less explicit than the one we found here. Furthermore, one could also consider the problem of relaxing the $C^2(\mathbb{R})$ hypothesis to include $C^1(\mathbb{R})$ or just continuous activations, like the famous ReLU function (i.e. $\text{ReLU}(x) = \max\{0, x\}$) which is excluded from our analysis. Some results in this direction can be found in Eldan et al. (2021), though using Rademacher weights for the hidden layer. In this regard, we try to derive a specific bound for the ReLU function applying Theorem 2.2 to a sequence of smooth approximating functions and then passing to the limit. In particular, we approximated the ReLU function with $G(m, x) := m^{-1} \log(1 + e^{mx})$ for $m \geq 1$ and

applied Theorem 2.2 to a generic $G(m, x)$ using the 1-Wasserstein distance and obtained a bound dependent on $m$. Then, the idea would have been to take the limit of this bound for $m \to \infty$ and hopefully obtain a non-trivial bound, but that was not the case as the limit exploded. The same outcome is found using the SAU approximating sequence, i.e.

$$H(m, x) := \frac{1}{m\sqrt{2\pi}} e^{-\frac{1}{2}m^2 x^2} + \frac{x}{2} + \frac{x}{2} \operatorname{erf}\left(\frac{mx}{\sqrt{2}}\right),$$

where $\operatorname{erf}(\cdot)$ denotes the error function. This fact indicates the impossibility to apply the results of Vidotto (2020) in the context of continuous activation functions as the ReLU function, and the necessity to come up with new results on second-order Poincaré inequalities to fill this gap. These results would not be trivial after all, since Theorem A.2 needs each $F_1, \ldots, F_d$ to be in $\mathbb{D}^{2,4}$, and so two degrees of smoothness are required. This is not "the fault" of Vidotto (2020), but it is due to the intrinsic character of the equation $f''(x) - xf'(x) = h(x) - Eh(Z)$ with $Z \sim \mathcal{N}(0, 1)$ in dimension $p \geq 2$.

## Acknowledgements

The authors are very grateful to three Referees for their comments and suggestions that improved remarkably the paper.

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

# A  Second-order Poincaré inequality for functionals of Gaussian fields

We present a brief overview of the main results of Vidotto (2020), of which Theorem 2.2 and Theorem 2.3 are special cases for random variables in $\mathbb{R}^d$. The main results of Vidotto (2020) improve on previous results of Nourdin et al. (2009), and such an improvement is obtained by using the Mehler representation of the Ornstein–Uhlenbeck semigroup, which was exploited in Last et al. (2016) to obtain second-order Poincaré inequalities for Poisson functionals. According to the Mehler formula, if $F \in L^1$, $X'$ is an independent copy of a random variable $X$, with $X$ and $X'$ being defined on the product probability space $(\Omega \times \Omega', \mathscr{F} \otimes \mathscr{F}', \mathbb{P} \times \mathbb{P}')$, and $P_t$ is the infinitesimal generator of the Ornstein–Uhlenbeck process then

$$P_t F = E\left[ f\left( e^{-t} X + \sqrt{1 - e^{-2t}} X' \right) \mid X \right], \quad t \geq 0.$$

Before stating Vidotto (2020, Theorem 2.1), it is useful to introduce some notation and definitions from Gaussian analysis and Malliavin calculus. We recall that an isonormal Gaussian process $X = \{X(h) : h \in H\}$ over $H = L^2(A, \mathscr{B}(A), \mu)$, where $(A, \mathscr{B}(A))$ is a Polish space endowed with its Borel $\sigma$-field and $\mu$ is a positive, $\sigma$-finite and non-atomic measure, is a centered Gaussian family defined on $(\Omega, \mathscr{F}, \mathbb{P})$ such that $E[X(h)X(g)] = \langle g, h \rangle_H$ for every $h, g \in H$. We denote by $L^2(\Omega; H)$ the set of $H$-valued random variables $Y$ satisfying $\mathbb{E}[\|Y\|_H^2] < \infty$. Furthermore, if $\mathcal{S}$ denotes the set of random variables of the form

$$F = f\left( X(\phi_1), \dots, X(\phi_m) \right),$$

where $f : \mathbb{R}^m \to \mathbb{R}$ is a $C^\infty$-function such that $f$ and its partial derivatives have at most polynomial growth at infinity, and $\phi_i \in H$, for $i = 1, \dots, m$, the Malliavin derivative of $F$ is the element of $L^2(\Omega; H)$ defined by

$$DF = \sum_{i=1}^m \frac{\partial f}{\partial x_i} \left( X(\phi_1), \dots, X(\phi_m) \right) \phi_i.$$

Moreover, in analogy with $DF$, the second Malliavin derivative of $F$ is the element of $L^2(\Omega; H^{\odot})$ defined by

$$D^2 F = \sum_{i,j=1}^m \frac{\partial^2 f}{\partial x_i \partial x_j} \left( X(\phi_1), \dots, X(\phi_m) \right) \phi_i \phi_j,$$

where $H^{\odot 2}$ is the second symmetric tensor power of $H$, so that $H^{\odot 2} = L_s^2\left( A^2, \mathscr{B}\left(A^2\right), \mu^2 \right)$ is the subspace of $L^2\left( A^2, \mathscr{B}\left(A^2\right), \mu^2 \right)$ whose elements are a.e. symmetric. Let us also define the Sobolev spaces $\mathbb{D}^{\alpha,p}$, $p \geq 1$, $\alpha = 1, 2$, which are defined as the closure of $\mathcal{S}$ with respect to the norms

$$\|F\|_{\mathbb{D}^{\alpha,p}} = \left( E\left[ |F|^p \right] + E\left[ \|DF\|_H^p + E\left[ \|D^2 F\|_{H^{\otimes 2}}^p \right] \mathbb{1}_{\{\alpha=2\}} \right] \right)^{1/p}.$$

In particular, the Sobolev space $\mathbb{D}^{\alpha,p}$ is typically referred to as the domain of $D^\alpha$ in $L^p(\Omega)$. Finally, for every $1 \leq m \leq n$, every $r = 1, \dots, m$, every $f \in L^2\left( A^m, \mathscr{B}\left(A^m\right), \mu^m \right)$ and every $g \in L^2\left( A^n, \mathscr{B}\left(A^n\right), \mu^n \right)$, the $r$-th contraction $f \otimes_r g : A^{n+m-2r} \to \mathbb{R}$ is defined to be the following function:

$$f \otimes_r g (y_1, \dots, y_{n+m-2r}) = \int_{A^r} f(x_1, \dots, x_r, y_1, \dots, y_{m-r})$$
$$\times g(x_1, \dots, x_r, y_{m-r+1}, \dots, y_{m+n-2r}) \, d\mu(x_1) \cdots d\mu(x_r).$$

Now, we can state Vidotto (2020, Theorem 2.1), which provides a second-order Poincaré inequality for a suitable class of functionals of Gaussian fields. For random variables in $\mathbb{R}^d$, the next theorem reduces to Theorem 2.2.

**Theorem A.1** (Vidotto (2020), Theorem 2.1). *Let $F \in \mathbb{D}^{2,4}$ be such that $E[F] = 0$ and $E\left[ F^2 \right] = \sigma^2$, and let $N \sim \mathcal{N}\left( 0, \sigma^2 \right)$; then,*

$$d_M(F, N) \leq c_M \left( \int_{A \times A} \left\{ E\left[ \left( \left( D^2 F \otimes_1 D^2 F \right)(x, y) \right)^2 \right] \right\}^{1/2} \right.$$
$$\left. \times \left\{ E\left[ (DF(x)DF(y))^2 \right] \right\}^{1/2} d\mu(x) d\mu(y) \right)^{1/2}$$

*where $M \in \{TV, KS, W_1\}$ and $c_{TV} = \frac{4}{\sigma^2}, c_{KS} = \frac{2}{\sigma^2}, c_{W_1} = \sqrt{\frac{8}{\sigma^2 \pi}}$.*

The novelty of Theorem A.1 lies in the fact that the upper bound is directly computable, making the approach of Vidotto (2020) very appealing for concrete applications of the Gaussian approximation. In particular, Theorem A.1 improves over previous results of Chatterjee (2009) and Nourdin et al. (2009). Now, we can state Vidotto (2020, Theorem 2.3), which provides a generalization of Theorem A.1 to multidimensional functionals. For random variables in $\mathbb{R}^d$, the next theorem reduces to Theorem 2.3.

**Theorem A.2** (Vidotto (2020), Theorem 2.3). *Let* $F = [F_1 \ldots F_p]$*, where, for each* $i = 1, \ldots, p, F_i \in \mathbb{D}^{2,4}$ *is such that* $E[F_i] = 0$ *and* $E[F_iF_j] = c_{ij}$*, with* $C = \{c_{ij}\}_{i,j=1,\ldots,p}$ *a symmetric and positive definite matrix. Let* $N \sim \mathcal{N}(0, C)$*, then we have that* $d_{W_1}(F, N) \leq 2\sqrt{p} \left\|C^{-1}\right\|_{op} \|C\|_{op} \times$

$$\sqrt{\sum_{i,k=1}^{p} \int_{A \times A} \left\{ E\left[ \left( \left(D^2 F_i \otimes_1 D^2 F_i\right)(x,y)\right)^2 \right] \right\}^{1/2} \left\{ E\left[ \left(DF_k(x)DF_k(y)\right)^2 \right] \right\}^{1/2} \mathrm{d}\mu(x)\mathrm{d}\mu(y)}.$$

## B   Proof of Theorem 3.1

To apply Theorem 2.2, we start by computing some first and second order partial derivatives. That is,

$$\begin{cases} \frac{\partial F}{\partial w_j} = n^{-1/2}\tau(w_j^{(0)}) \\[2mm] \frac{\partial F}{\partial w_j^{(0)}} = n^{-1/2}w_j\tau'(w_j^{(0)}) \\[2mm] \nabla^2_{w_j, w_i} F = 0 \\[2mm] \nabla^2_{w_j, w_i^{(0)}} F = n^{-1/2}\tau'(w_j^{(0)})\delta_{ij} \\[2mm] \nabla^2_{w_j^{(0)}, w_i^{(0)}} F = n^{-1/2}w_j\tau''(w_j^{(0)})\delta_{ij} \end{cases}$$

with $i, j = 1 \ldots n$. Then, by a direct application of Theorem 2.2, we obtain the following preliminary estimate

$$\begin{aligned} d_M(F, N) \leq c_M \Bigg\{ &\sum_{j=1}^{n} 2 \left\{ \mathbb{E}\left[ \left(\langle \nabla^2_{w_j,\cdot} F, \nabla^2_{w_j^{(0)},\cdot} F\rangle\right)^2 \right] \mathbb{E}\left[ \left(\frac{\partial F}{\partial w_j}\frac{\partial F}{\partial w_j^{(0)}}\right)^2 \right] \right\}^{1/2} \\ &+ \left\{ \mathbb{E}\left[ \left(\langle \nabla^2_{w_j,\cdot} F, \nabla^2_{w_j,\cdot} F\rangle\right)^2 \right] \mathbb{E}\left[ \left(\frac{\partial F}{\partial w_j}\frac{\partial F}{\partial w_j}\right)^2 \right] \right\}^{1/2} \\ &+ \left\{ \mathbb{E}\left[ \left(\langle \nabla^2_{w_j^{(0)},\cdot} F, \nabla^2_{w_j^{(0)},\cdot} F\rangle\right)^2 \right] \mathbb{E}\left[ \left(\frac{\partial F}{\partial w_j^{(0)}}\frac{\partial F}{\partial w_j^{(0)}}\right)^2 \right] \right\}^{1/2} \Bigg\}^{1/2}, \end{aligned}$$

which can be further developed. In particular, we can write the right-hand side of the previous estimate as

$$\begin{aligned} c_M \Bigg\{ &\sum_{j=1}^{n} 2 \left\{ \mathbb{E}\left[ \left(\frac{1}{n}w_j\tau'\left(w_j^{(0)}\right)\tau''\left(w_j^{(0)}\right)\right)^2 \right] \mathbb{E}\left[ \left(\frac{1}{n}w_j\tau\left(w_j^{(0)}\right)\tau'\left(w_j^{(0)}\right)\right)^2 \right] \right\}^{1/2} \\ &+ \left\{ \mathbb{E}\left[ \left(\frac{1}{\sqrt{n}}\tau'\left(w_j^{(0)}\right)\right)^4 \right] \mathbb{E}\left[ \left(\frac{1}{\sqrt{n}}\tau\left(w_j^{(0)}\right)\right)^4 \right] \right\}^{1/2} \\ &+ \left\{ \mathbb{E}\left[ \left(\frac{1}{n}\left\{\tau'\left(w_j^{(0)}\right)\right\}^2 + \frac{1}{n}w_j^2\left\{\tau''\left(w_j^{(0)}\right)\right\}^2\right)^2 \right] \mathbb{E}\left[ \left(\frac{1}{\sqrt{n}}w_j\tau'\left(w_j^{(0)}\right)\right)^4 \right] \right\}^{1/2} \Bigg\}^{1/2} \end{aligned}$$

$$
\overset{(\mathbb{E}[w_j^2]=1)}{=} \frac{c_M}{n} \left\{ \sum_{j=1}^{n} 2 \left\{ \mathbb{E}\left[ \left( \tau'\left(w_j^{(0)}\right) \tau''\left(w_j^{(0)}\right) \right)^2 \right] \mathbb{E}\left[ \left( \tau\left(w_j^{(0)}\right) \tau'\left(w_j^{(0)}\right) \right)^2 \right] \right\}^{1/2} \right.
$$

$$
+ \left\{ \mathbb{E}\left[ \left( \tau'\left(w_j^{(0)}\right) \right)^4 \right] \mathbb{E}\left[ \left( \tau\left(w_j^{(0)}\right) \right)^4 \right] \right\}^{1/2}
$$

$$
\left. + \left\{ \mathbb{E}\left[ \left( \left\{ \tau'\left(w_j^{(0)}\right) \right\}^2 + w_j^2 \left\{ \tau''\left(w_j^{(0)}\right) \right\}^2 \right)^2 \right] \mathbb{E}\left[ \left( w_j \tau'\left(w_j^{(0)}\right) \right)^4 \right] \right\}^{1/2} \right\}^{1/2}
$$

$$
\overset{(iid)}{=} \frac{c_M}{\sqrt{n}} \left\{ 2 \left\{ \mathbb{E}\left[ (\tau'(Z)\tau''(Z))^2 \right] \mathbb{E}\left[ (\tau(Z)\tau'(Z))^2 \right] \right\}^{1/2} \right.
$$

$$
+ \left\{ \mathbb{E}\left[ (\tau'(Z))^4 \right] \mathbb{E}\left[ (\tau(Z))^4 \right] \right\}^{1/2}
$$

$$
\left. + \left\{ \mathbb{E}\left[ \left( \{\tau'(Z)\}^2 + w_j^2 \{\tau''(Z)\}^2 \right)^2 \right] \mathbb{E}\left[ (w_j \tau'(Z))^4 \right] \right\}^{1/2} \right\}^{1/2}
$$

$$
\overset{(iid)}{=} \frac{c_M}{\sqrt{n}} \left\{ 2 \left\{ \mathbb{E}\left[ (\tau'(Z)\tau''(Z))^2 \right] \mathbb{E}\left[ (\tau(Z)\tau'(Z))^2 \right] \right\}^{1/2} \right.
$$

$$
+ \left\{ \mathbb{E}\left[ (\tau'(Z))^4 \right] \mathbb{E}\left[ (\tau(Z))^4 \right] \right\}^{1/2}
$$

$$
\left. + \left\{ \mathbb{E}\left[ \left( \{\tau'(Z)\}^2 + w_j^2 \{\tau''(Z)\}^2 \right)^2 \right] \mathbb{E}\left[ (w_j \tau'(Z))^4 \right] \right\}^{1/2} \right\}^{1/2}
$$

$$
= \frac{c_M}{\sqrt{n}} \left\{ 2 \left\{ \mathbb{E}\left[ (\tau'(Z)\tau''(Z))^2 \right] \mathbb{E}\left[ (\tau(Z)\tau'(Z))^2 \right] \right\}^{1/2} \right.
$$

$$
+ \left\{ \mathbb{E}\left[ (\tau'(Z))^4 \right] \mathbb{E}\left[ (\tau(Z))^4 \right] \right\}^{1/2}
$$

$$
\left. + \left\{ \left( \mathbb{E}\left[ \{\tau'(Z)\}^4 \right] + 2\mathbb{E}\left[ \{\tau'(Z)\}^2 \{\tau''(Z)\}^2 \right] + 3\mathbb{E}\left[ \{\tau''(Z)\}^4 \right] \right) 3\mathbb{E}\left[ \{\tau'(Z)\}^4 \right] \right\}^{1/2} \right\}^{1/2}
$$

$$
= \frac{c_M}{\sqrt{n}} \left\{ 2 \left\{ \mathbb{E}\left[ |\tau'(Z)|^2 |\tau''(Z)|^2 \right] \mathbb{E}\left[ |\tau(Z)|^2 |\tau'(Z)|^2 \right] \right\}^{1/2} \right.
$$

$$
+ \left\{ \mathbb{E}\left[ |\tau'(Z)|^4 \right] \mathbb{E}\left[ |\tau(Z)|^4 \right] \right\}^{1/2}
$$

$$
\left. + \left\{ \left( \mathbb{E}\left[ |\tau'(Z)|^4 \right] + 2\mathbb{E}\left[ |\tau'(Z)|^2 |\tau''(Z)|^2 \right] + 3\mathbb{E}\left[ |\tau''(Z)|^4 \right] \right) 3\mathbb{E}\left[ |\tau'(Z)|^4 \right] \right\}^{1/2} \right\}^{1/2},
$$

where $Z \sim \mathcal{N}(0,1)$. Now, since $\tau$ is polynomially bounded and the square root is an increasing function,

$$
d_M(F, N) \leq \frac{c_M}{\sqrt{n}} \left\{ 2 \left\{ \mathbb{E}\left[ (\alpha + \beta|Z|^\gamma)^4 \right] \mathbb{E}\left[ (\alpha + \beta|Z|^\gamma)^4 \right] \right\}^{1/2} \right.
$$

$$
+ \left\{ \mathbb{E}\left[ (\alpha + \beta|Z|^\gamma)^4 \right] \mathbb{E}\left[ (\alpha + \beta|Z|^\gamma)^4 \right] \right\}^{1/2}
$$

$$
\left. + \left\{ 18\mathbb{E}\left[ (\alpha + \beta|Z|^\gamma)^4 \right] \mathbb{E}\left[ (\alpha + \beta|Z|^\gamma)^4 \right] \right\}^{1/2} \right\}
$$

$$
= \frac{c_M}{\sqrt{n}} \sqrt{3\sqrt{2} + 3} \left\{ \mathbb{E}\left[ (\alpha + \beta|Z|^\gamma)^4 \right] \right\}^{1/2}
$$

$$
= \frac{c_M}{\sqrt{n}} \sqrt{3(1 + \sqrt{2})} \| \alpha + \beta|Z|^\gamma \|_{L_4}^2,
$$

where $Z \sim \mathcal{N}(0,1)$.

## C  Proof of Theorem 3.2

As stated in the main body, we will make use of the fact that

$$F \stackrel{d}{=} \tilde{F} := n^{-1/2}\sigma_w \sum_{j=1}^{n} w_j \tau \left(\Gamma \cdot Y_j\right) + \sigma_b \cdot b,$$

where $\Gamma = \sigma_w^2 \|x\|^2 + \sigma_b^2$. First, it is easy to see that $\mathbb{E}[F] = 0$ and that

$$\sigma^2 = \text{Var}[F] = \text{Var}[\tilde{F}] = \sigma_w^2 \mathbb{E}_{Z \sim \mathcal{N}(0,1)} \left[\tau^2 \left(\Gamma Z\right)\right] + \sigma_b^2.$$

Then we have that $d_M(F, N) = d_M(\tilde{F}, N)$, where $N \sim \mathcal{N}(0, \sigma^2)$, hence it is enough to apply Theorem 2.2 to $\tilde{F}$. To this aim, we compute again the gradient and the Hessian of $\tilde{F}$, noticing that the only difference with the Shallow case lies in the presence of an extra factor $\sigma_w$ in front of the sum, an extra factor of $\Gamma$ inside the activation and the bias term $\sigma_b^2 b$:

$$\begin{cases} \frac{\partial \tilde{F}}{\partial b} = \sigma_b \\[2mm] \frac{\partial \tilde{F}}{\partial w_j} = n^{-1/2}\sigma_w \cdot \tau\left(\Gamma Y_j\right) \\[2mm] \frac{\partial \tilde{F}}{\partial Y_j} = n^{-1/2}\sigma_w \Gamma \cdot w_j \cdot \tau'\left(\Gamma Y_j\right) \\[2mm] \nabla_{b,\cdot}^2 \tilde{F} = 0 \\[2mm] \nabla_{w_j, w_i}^2 \tilde{F} = 0 \\[2mm] \nabla_{w_j, Y_i}^2 \tilde{F} = n^{-1/2}\sigma_w \Gamma \cdot \tau'\left(\Gamma Y_j\right)\delta_{ij} \\[2mm] \nabla_{Y_j, Y_i}^2 \tilde{F} = n^{-1/2}\sigma_w \Gamma^2 \cdot w_j \cdot \tau''\left(\Gamma Y_j\right)\delta_{ij} \end{cases}$$

It is interesting to notice that since the row of the Hessian corresponding to the bias term $b$ contains all zeros, then the bound given by Theorem 2.2 is exactly the same as the one at the beginning of the proof of Theorem 3.1, with the only difference that now the expectations depend also on $\Gamma$ and $\sigma_w$. More precisely, we have that

$$d_M\left(F, N\right) = d_M\left(\tilde{F}, N\right) \leq$$

$$\leq c_M \Bigg\{ \sum_{j=1}^{n} 2 \left\{ \mathbb{E}\left[\left(\langle \nabla_{w_j,\cdot}^2 \tilde{F}, \nabla_{Y_j,\cdot}^2 \tilde{F}\rangle\right)^2\right] \cdot \mathbb{E}\left[\left(\frac{\partial \tilde{F}}{\partial w_j} \cdot \frac{\partial \tilde{F}}{\partial Y_j}\right)^2\right] \right\}^{1/2}$$

$$+ \left\{ \mathbb{E}\left[\left(\langle \nabla_{w_j,\cdot}^2 \tilde{F}, \nabla_{w_j,\cdot}^2 \tilde{F}\rangle\right)^2\right] \cdot \mathbb{E}\left[\left(\frac{\partial \tilde{F}}{\partial w_j} \cdot \frac{\partial \tilde{F}}{\partial w_j}\right)^2\right] \right\}^{1/2}$$

$$+ \left\{ \mathbb{E}\left[\left(\langle \nabla_{Y_j,\cdot}^2 \tilde{F}, \nabla_{Y_j,\cdot}^2 \tilde{F}\rangle\right)^2\right] \cdot \mathbb{E}\left[\left(\frac{\partial \tilde{F}}{\partial Y_j} \cdot \frac{\partial \tilde{F}}{\partial Y_j}\right)^2\right] \right\}^{1/2} \Bigg\}^{1/2}$$

$$= c_M \Bigg\{ \sum_{j=1}^{n} 2 \left\{ \mathbb{E}\left[\left(\frac{1}{n}\sigma_w^2 \Gamma^3 w_j \tau'\left(\Gamma Y_j\right) \tau''\left(\Gamma Y_j\right)\right)^2\right] \cdot \mathbb{E}\left[\left(\frac{1}{n}\sigma_w^2 \Gamma w_j \tau\left(\Gamma Y_j\right)\tau'\left(\Gamma Y_j\right)\right)^2\right] \right\}^{1/2}$$

$$+ \left\{ \mathbb{E}\left[\left(\frac{1}{\sqrt{n}}\sigma_w \Gamma \tau'\left(\Gamma Y_j\right)\right)^4\right] \cdot \mathbb{E}\left[\left(\frac{1}{\sqrt{n}}\sigma_w \tau\left(\Gamma Y_j\right)\right)^4\right] \right\}^{1/2}$$

$$+ \left\{ \mathbb{E}\left[\left(\frac{1}{n}\sigma_w^2 \Gamma^2 \{\tau'\left(\Gamma Y_j\right)\}^2 + \frac{1}{n}\sigma_w^2 \Gamma^4 w_j^2 \{\tau''\left(\Gamma Y_j\right)\}^2\right)^2\right] \right.$$

$$\times \mathbb{E}\left[\left(\frac{1}{\sqrt{n}}\sigma_w\Gamma w_j\tau'\left(\Gamma Y_j\right)\right)^4\right]\bigg\}^{1/2}\bigg\}^{1/2}$$

$$\overset{\mathbb{E}w_j^2=1}{=}\frac{c_M}{n}\sigma_w^2\bigg\{\sum_{j=1}^n 2\Gamma^4\left\{\mathbb{E}\left[\left(\tau'\left(\Gamma Y_j\right)\tau''\left(\Gamma Y_j\right)\right)^2\right]\cdot\mathbb{E}\left[\left(\tau\left(\Gamma Y_j\right)\tau'\left(\Gamma Y_j\right)\right)^2\right]\right\}^{1/2}$$

$$+\Gamma^2\left\{\mathbb{E}\left[\left(\tau'\left(\Gamma Y_j\right)\right)^4\right]\cdot\mathbb{E}\left[\left(\tau\left(\Gamma Y_j\right)\right)^4\right]\right\}^{1/2}$$

$$+\left\{\mathbb{E}\left[\left(\Gamma^2\left\{\tau'\left(\Gamma Y_j\right)\right\}^2+\Gamma^4 w_j^2\left\{\tau''\left(\Gamma Y_j\right)\right\}^2\right)^2\right]\cdot\mathbb{E}\left[\left(\Gamma w_j\tau'\left(\Gamma Y_j\right)\right)^4\right]\right\}^{1/2}\bigg\}^{1/2}\bigg\}$$

$$\overset{iid}{=}\frac{c_M}{\sqrt{n}}\sigma_w^2\bigg\{2\Gamma^4\left\{\mathbb{E}\left[\left(\tau'\left(\Gamma Z\right)\tau''\left(\Gamma Z\right)\right)^2\right]\cdot\mathbb{E}\left[\left(\tau\left(\Gamma Z\right)\tau'\left(\Gamma Z\right)\right)^2\right]\right\}^{1/2}$$

$$+\Gamma^2\left\{\mathbb{E}\left[\left(\tau'\left(\Gamma Z\right)\right)^4\right]\cdot\mathbb{E}\left[\left(\tau\left(\Gamma Z\right)\right)^4\right]\right\}^{1/2}$$

$$+\left\{\mathbb{E}\left[\left(\Gamma^2\left\{\tau'\left(\Gamma Z\right)\right\}^2+\Gamma^4 w_j^2\left\{\tau''\left(\Gamma Z\right)\right\}^2\right)^2\right]\cdot\mathbb{E}\left[\left(\Gamma w_j\tau'\left(\Gamma Z\right)\right)^4\right]\right\}^{1/2}\bigg\}^{1/2}$$

$$=\frac{c_M}{\sqrt{n}}\sigma_w^2\bigg\{2\Gamma^4\left\{\mathbb{E}\left[\left(\tau'\left(\Gamma Z\right)\tau''\left(\Gamma Z\right)\right)^2\right]\cdot\mathbb{E}\left[\left(\tau\left(\Gamma Z\right)\tau'\left(\Gamma Z\right)\right)^2\right]\right\}^{1/2}$$

$$+\Gamma^2\left\{\mathbb{E}\left[\left(\tau'\left(\Gamma Z\right)\right)^4\right]\cdot\mathbb{E}\left[\left(\tau\left(\Gamma Z\right)\right)^4\right]\right\}^{1/2}$$

$$+\left\{\left(\Gamma^4\mathbb{E}\left[\left\{\tau'\left(\Gamma Z\right)\right\}^4\right]+2\Gamma^6\mathbb{E}\left[\left\{\tau'\left(\Gamma Z\right)\right\}^2\left\{\tau''\left(\Gamma Z\right)\right\}^2\right]+3\Gamma^8\mathbb{E}\left[\left\{\tau''\left(\Gamma Z\right)\right\}^4\right]\right)\right.$$

$$\left.\times 3\Gamma^4\cdot\mathbb{E}\left[\left\{\tau'\left(\Gamma Z\right)\right\}^4\right]\right\}^{1/2}\bigg\}^{1/2}$$

$$=\frac{c_M}{\sqrt{n}}\sigma_w^2\bigg\{2\Gamma^4\left\{\mathbb{E}\left[|\tau'\left(\Gamma Z\right)|^2|\tau''\left(\Gamma Z\right)|^2\right]\cdot\mathbb{E}\left[|\tau\left(\Gamma Z\right)|^2|\tau'\left(\Gamma Z\right)|^2\right]\right\}^{1/2}$$

$$+\Gamma^2\left\{\mathbb{E}\left[|\tau'\left(\Gamma Z\right)|^4\right]\cdot\mathbb{E}\left[|\tau\left(\Gamma Z\right)|^4\right]\right\}^{1/2}$$

$$+\left\{\left(\Gamma^4\mathbb{E}\left[|\tau'\left(\Gamma Z\right)|^4\right]+2\Gamma^6\cdot\mathbb{E}\left[|\tau'\left(\Gamma Z\right)|^2|\tau''\left(\Gamma Z\right)|^2\right]+3\Gamma^8\cdot\mathbb{E}\left[|\tau''\left(\Gamma Z\right)|^4\right]\right)\right.$$

$$\left.\times 3\Gamma^4\cdot\mathbb{E}\left[|\tau'\left(\Gamma Z\right)|^4\right]\right\}^{1/2}\bigg\}$$

where $Z\sim\mathcal{N}(0,1)$. But since $\tau$ is polynomially bounded and the square root is an increasing function, we can bound this expression by

$$\frac{c_M}{\sqrt{n}}\sigma_w^2\bigg\{2\Gamma^4\left\{\mathbb{E}\left[(\alpha+\beta|\Gamma Z|^\gamma)^4\right]\cdot\mathbb{E}\left[(\alpha+\beta|\Gamma Z|^\gamma)^4\right]\right\}^{1/2}$$

$$+\Gamma^2\left\{\mathbb{E}\left[(\alpha+\beta|\Gamma Z|^\gamma)^4\right]\cdot\mathbb{E}\left[(\alpha+\beta|\Gamma Z|^\gamma)^4\right]\right\}^{1/2}$$

$$+\Gamma^4\left\{\sqrt{3(1+2\Gamma^2+3\Gamma^4)}\cdot\mathbb{E}\left[(\alpha+\beta|\Gamma Z|^\gamma)^4\right]\cdot\mathbb{E}\left[(\alpha+\beta|\Gamma Z|^\gamma)^4\right]\right\}^{1/2}\bigg\}^{1/2}$$

$$=\frac{c_M}{\sqrt{n}}\sigma_w^2\sqrt{\Gamma^2+\Gamma^4(2+\sqrt{3(1+2\Gamma^2+3\Gamma^4)})}\left\{\mathbb{E}\left[(\alpha+\beta|\Gamma Z|^\gamma)^4\right]\right\}^{1/2}$$

$$= \frac{c_M}{\sqrt{n}} \sigma_w^2 \sqrt{\Gamma^2 + \Gamma^4 (2 + \sqrt{3(1 + 2\Gamma^2 + 3\Gamma^4)})} \cdot \|\alpha + \beta|\Gamma Z|^\gamma\|_{L^4}^2,$$

where $Z \sim \mathcal{N}(0, 1)$.

## D   Proof of Theorem 3.3

The proof is based on Theorem 2.3. Recall that

$$F_i := \frac{1}{n^{1/2}} \sigma_w \sum_{j=1}^{n} w_j \tau(\sigma_w \langle w_j^{(0)}, \boldsymbol{x_i} \rangle + \sigma_b b_j^{(0)}) + \sigma_b b.$$

Since $F_1, \ldots, F_p$ are functions of the iid standard normal random variables $\{w_j, w_{jl}^{(0)}, b_j^{(0)}, b : j = 1, \ldots, n, l = 1, \ldots, d\}$, then we can apply Theorem 2.3 to the random vector $F = [F_1 \; \cdots \; F_p]$. The upper bound in (8) depends on the first and second derivatives of the $F_i$'s with respect to all their arguments. However, the derivatives with respect to $b$ give no contributions, since, for every $i = 1, \ldots, p$, $\nabla_{b,\cdot}^2 F_i$ is the zero vector. Moreover, the terms $w_j \tau(\sigma_w \langle w_j^{(0)}, \boldsymbol{x_i} \rangle + \sigma_b b_j^{(0)})$ are iid, across $j$, and give the same contribution to the upper bound. Hence, we can write that

$$d_{W_1}(F, N) \le 2\sigma_w^2 \sqrt{\frac{p}{n}} \|C^{-1}\|_2 \|C\|_2 \sqrt{\sum_{i,k=1}^{p} D_{ik}},$$

where

$$D_{ik} = \sum_{l,m} \left\{ \mathbb{E}\left[ (\langle \nabla_{l,\cdot}^2 \tilde{F}_i, \nabla_{m,\cdot}^2 \tilde{F}_i \rangle)^2 \right] \right\}^{1/2} \left\{ \mathbb{E}\left[ (\nabla_l \tilde{F}_k \nabla_m \tilde{F}_k)^2 \right] \right\}^{1/2},$$

where

$$[\tilde{F}_1 \; \ldots \; \tilde{F}_p] \stackrel{d}{=} [w_j \tau(\sigma_w \langle w_j^{(0)}, \boldsymbol{x_1} \rangle + \sigma_b b_j^{(0)}) \; \ldots \; w_j \tau(\sigma_w \langle w_j^{(0)}, \boldsymbol{x_p} \rangle + \sigma_b b_j^{(0)})],$$

and $\nabla_l, \nabla_m, \nabla_{l,\cdot}^2$ and $\nabla_{m,\cdot}^2$ denote the derivatives with respect to all the arguments. We can represent $\tilde{F}_i$ as

$$\tilde{F}_i = w \cdot \tau(Y_i),$$

where $Y_i := \langle \tilde{w}^{(0)}, \tilde{\boldsymbol{x}_i} \rangle = \sum_{s=1}^{d} \tilde{w}_s^{(0)} \tilde{x}_{is}$, with $\tilde{\boldsymbol{x}_i} := [\sigma_w \boldsymbol{x}_i^T, \sigma_b]^T$, $\tilde{w}^{(0)} := [w^{(0)T}, b^{(0)}]^T$, and $w, \tilde{w}_1^{(0)}, \ldots, \tilde{w}_d^{(0)}, b^{(0)}$ iid standard normal random variables. The gradient and the Hessian of $\tilde{F}$ with respect to the parameters $w$ and $\tilde{w}_s^{(0)}$ are

$$\begin{cases} \frac{\partial \tilde{F}_i}{\partial w} = \tau(Y_i) \\[2mm] \frac{\partial \tilde{F}_i}{\partial w_s^{(0)}} = w\tau'(Y_i)\tilde{x}_{is} \\[2mm] \nabla_{w,w}^2 \tilde{F}_i = 0 \\[2mm] \nabla_{w,\tilde{w}_s^{(0)}}^2 \tilde{F}_i = \tau'(Y_i)\tilde{x}_{is} \\[2mm] \nabla_{\tilde{w}_s^{(0)}, \tilde{w}_t^{(0)}}^2 \tilde{F}_i = w\tau''(Y_i)\tilde{x}_{is}\tilde{x}_{it}. \end{cases}$$

This implies that

$$D_{ik} = \left\{ \mathbb{E}\left[ \left( \sum_{s=1}^{d} \nabla_{w,\tilde{w}_s^{(0)}}^2 \tilde{F}_i \cdot \nabla_{w,\tilde{w}_s^{(0)}}^2 \tilde{F}_i \right)^2 \right] \right\}^{1/2} \left\{ \mathbb{E}\left[ \left( \frac{\partial \tilde{F}_k}{\partial w} \cdot \frac{\partial \tilde{F}_k}{\partial w} \right)^2 \right] \right\}^{1/2}$$

$$+ \sum_{j,j'=1}^{d} \left\{ \mathbb{E}\left[ \left( \nabla^2_{w,\tilde{w}_j^{(0)}} \tilde{F}_i \cdot \nabla^2_{w,\tilde{w}_{j'}^{(0)}} \tilde{F}_i + \sum_{s=1}^{d} \nabla^2_{\tilde{w}_j^{(0)},\tilde{w}_s^{(0)}} \tilde{F}_i \cdot \nabla^2_{\tilde{w}_{j'}^{(0)},\tilde{w}_s^{(0)}} \tilde{F}_i \right)^2 \right] \right\}^{1/2}$$

$$\times \left\{ \mathbb{E}\left[ \left( \frac{\partial \tilde{F}_k}{\partial \tilde{w}_j^{(0)}} \cdot \frac{\partial \tilde{F}_k}{\partial \tilde{w}_{j'}^{(0)}} \right)^2 \right] \right\}^{1/2}$$

$$+ 2\sum_{j=1}^{d} \left\{ \mathbb{E}\left[ \left( \sum_{s=1}^{d} \nabla^2_{w,\tilde{w}_s^{(0)}} \tilde{F}_i \cdot \nabla^2_{\tilde{w}_j^{(0)},\tilde{w}_s^{(0)}} \tilde{F}_i \right)^2 \right] \right\}^{1/2} \left\{ \mathbb{E}\left[ \left( \frac{\partial \tilde{F}_k}{\partial w} \cdot \frac{\partial \tilde{F}_k}{\partial \tilde{w}_j^{(0)}} \right)^2 \right] \right\}^{1/2}$$

$$= \left\{ \mathbb{E}\left[ \left( \sum_{s=1}^{d} \tau'(Y_i)^2 \tilde{x}_{is}^2 \right)^2 \right] \right\}^{1/2} \left\{ \mathbb{E}\left[ (\tau(Y_k))^4 \right] \right\}^{1/2}$$

$$+ \sum_{j,j'=1}^{d} \left\{ \mathbb{E}\left[ \left( \tau'(Y_i)^2 \tilde{x}_{ij}\tilde{x}_{ij'} + \sum_{s=1}^{d} w^2 \tau''(Y_i)^2 \tilde{x}_{ij}\tilde{x}_{ij'}\tilde{x}_{is}^2 \right)^2 \right] \right\}^{1/2} \left\{ \mathbb{E}\left[ (w^2 \tau'(Y_k)^2 \tilde{x}_{kj}\tilde{x}_{kj'})^2 \right] \right\}^{1/2}$$

$$+ 2\sum_{j=1}^{d} \left\{ \mathbb{E}\left[ \left( \sum_{s=1}^{d} \tau'(Y_i)\tilde{x}_{is} w \tau''(Y_i)\tilde{x}_{ij}\tilde{x}_{is} \right)^2 \right] \right\}^{1/2} \left\{ \mathbb{E}\left[ (\tau(Y_k) w \tau'(Y_k)\tilde{x}_{kj})^2 \right] \right\}^{1/2}$$

$$= ||\tilde{\boldsymbol{x}}_i||^2 ||\tau'(Y_i)||^2_{L_4} ||\tau(Y_k)||^2_{L_4}$$

$$+ \sum_{j,j'=1}^{d} |\tilde{x}_{ij}\tilde{x}_{ij'}| \left\{ \mathbb{E}\left[ \left( \tau'(Y_i)^2 + w^2 \tau''(Y_i)^2 ||\tilde{\boldsymbol{x}}_i||^2 \right)^2 \right] \right\}^{1/2} \sqrt{3} |\tilde{x}_{kj}\tilde{x}_{kj'}| ||\tau'(Y_k)||^2_{L^4}$$

$$+ 2\sum_{j=1}^{d} |\tilde{x}_{ij}||\tilde{x}_{kj}| ||\tilde{\boldsymbol{x}_i}||^2 \left\{ \mathbb{E}\left[ \left( \tau'(Y_i)\tau''(Y_i) \right)^2 \right] \right\}^{1/2} \left\{ \mathbb{E}\left[ \left( \tau(Y_k)\tau'(Y_k) \right)^2 \right] \right\}^{1/2}$$

$$= ||\tilde{\boldsymbol{x}}_i||^2 ||\tau'(Y_i)||^2_{L_4} ||\tau(Y_k)||^2_{L_4} + \sum_{j,j'=1}^{d} \sqrt{3} |\tilde{x}_{kj}\tilde{x}_{kj'}| |\tilde{x}_{ij}\tilde{x}_{ij'}| ||\tau'(Y_k)||^2_{L^4}$$

$$\times \left\{ ||\tau'(Y_i)||^4_{L_4} + 3||\tilde{\boldsymbol{x}_i}||^4 \left\| \tau''(Y_i) \right\|^4_{L_4} + 2||\tilde{\boldsymbol{x}_i}||^2 \left\| \tau'(Y_i)\tau''(Y_i) \right\|^2_{L_2} \right\}^{1/2}$$

$$+ 2\sum_{j=1}^{d} |\tilde{x}_{ij}||\tilde{x}_{kj}| ||\tilde{\boldsymbol{x}_i}||^2 \left\| \tau'(Y_i)\tau''(Y_i) \right\|_{L_2} \left\| \tau(Y_k)\tau'(Y_k) \right\|_{L_2}$$

$$= ||\tilde{\boldsymbol{x}}_i||^2 ||\tau'(Y_i)||^2_{L_4} ||\tau(Y_k)||^2_{L_4} + \sqrt{3} ||\tau'(Y_k)||^2_{L^4} \left( \sum_{j=1}^{d} |\tilde{x}_{ij}\tilde{x}_{kj}| \right)^2$$

$$\times \left\{ ||\tau'(Y_i)||^4_{L_4} + 3||\tilde{\boldsymbol{x}_i}||^4 \left\| \tau''(Y_i) \right\|^4_{L_4} + 2||\tilde{\boldsymbol{x}_i}||^2 \left\| \tau'(Y_i)\tau''(Y_i) \right\|^2_{L_2} \right\}^{1/2}$$

$$+ 2||\tilde{\boldsymbol{x}_i}||^2 \left\| \tau'(Y_i)\tau''(Y_i) \right\|_{L_2} \left\| \tau(Y_k)\tau'(Y_k) \right\|_{L_2} \left( \sum_{j=1}^{d} |\tilde{x}_{ij}||\tilde{x}_{kj}| \right)$$

$$\overset{\text{Holder ineq.}}{\leq} ||\tilde{\boldsymbol{x}}_i||^2 ||\tau'(Y_i)||^2_{L_4} ||\tau(Y_k)||^2_{L_4} + \sqrt{3} ||\tau'(Y_k)||^2_{L^4} \left( \sum_{j=1}^{d} |\tilde{x}_{ij}\tilde{x}_{kj}| \right)^2$$

$$\times \left\{ ||\tau'(Y_i)||^4_{L_4} + 3||\tilde{\boldsymbol{x}_i}||^4 \left\| \tau''(Y_i) \right\|^4_{L_4} + 2||\tilde{\boldsymbol{x}_i}||^2 \left\| \tau'(Y_i) \right\|^2_{L_4} \left\| \tau''(Y_i) \right\|^2_{L_4} \right\}^{1/2}$$

$$+ 2\|\tilde{\boldsymbol{x}_i}\|^2 \left\|\tau^{'}(Y_i)\right\|_{L_4} \left\|\tau^{''}(Y_i)\right\|_{L_4} \|\tau(Y_k)\|_{L_4} \left\|\tau^{'}(Y_k)\right\|_{L_4} \left(\sum_{j=1}^{d} |\tilde{x}_{ij}||\tilde{x}_{kj}|\right)$$

$$\overset{\text{polynom. bounded}}{\leq} \|\tilde{\boldsymbol{x}}_i\|^2 \|\alpha + \beta|Y_i|^\gamma\|_{L_4}^2 \|\alpha + \beta|Y_k|^\gamma\|_{L_4}^2$$

$$+ \sqrt{3} \left\{ (1 + 2\|\tilde{\boldsymbol{x}_i}\|^2 + 3\|\tilde{\boldsymbol{x}_i}\|^4)\|\alpha + \beta|Y_i|^\gamma\|_{L_4}^4 \right\}^{1/2} \|\alpha + \beta|Y_k|^\gamma\|_{L_4}^2 \left(\sum_{j=1}^{d} |\tilde{x}_{ij}\tilde{x}_{kj}|\right)^2$$

$$+ 2\|\tilde{\boldsymbol{x}_i}\|^2 \|\alpha + \beta|Y_i|^\gamma\|_{L_4}^2 \|\alpha + \beta|Y_k|^\gamma\|_{L_4}^2 \left(\sum_{j=1}^{d} |\tilde{x}_{ij}\tilde{x}_{kj}|\right)$$

$$= \left\{ \|\tilde{\boldsymbol{x}}_i\|^2 + \sqrt{3(1 + 2\|\tilde{\boldsymbol{x}_i}\|^2 + 3\|\tilde{\boldsymbol{x}_i}\|^4)} \left(\sum_{j=1}^{d} |\tilde{x}_{ij}\tilde{x}_{kj}|\right)^2 + 2\|\tilde{\boldsymbol{x}_i}\|^2 \left(\sum_{j=1}^{d} |\tilde{x}_{ij}\tilde{x}_{kj}|\right) \right\}$$

$$\times \|\alpha + \beta|Y_i|^\gamma\|_{L_4}^2 \|\alpha + \beta|Y_k|^\gamma\|_{L_4}^2.$$

Now, traducing everything back to the original variables $\{\boldsymbol{x}_i\}_{i\in[d]}$, we have that

$$\begin{cases} \sum_{j=1}^{d} |\tilde{x}_{ij}||\tilde{x}_{kj}| = \sigma_w^2 \sum_{j=1}^{d} |x_{ij}||x_{kj}| + \sigma_b^2 =: \Gamma_{ik} \\ \\ \|\tilde{\boldsymbol{x}}_i\|^2 = \sigma_w^2 \|\boldsymbol{x}_i\|^2 + \sigma_b^2 =: \Gamma_i^2. \end{cases}$$

Hence,

$$D_{ik} \leq (\Gamma_i^2 + \sqrt{3(1 + 2\Gamma_i^2 + 3\Gamma_i^4)}\Gamma_{ik}^2 + 2\Gamma_i^2\Gamma_{ik})\|\alpha + \beta|Y_i|^\gamma\|_{L_4}^2 \|\alpha + \beta|Y_k|^\gamma\|_{L_4}^2,$$

with $Y \sim \mathcal{N}(0, \sigma_b^2 \mathbf{X}^T\mathbf{X} + \sigma_b^2 \mathbf{1}\mathbf{1}^T)$. Summing over all possible $i, k = 1, \ldots, p$ and taking the square root leads to

$$d_{W_1}(F, N) \leq 2\sigma_w^2 \frac{\lambda_1(C)}{\lambda_p(C)} \sqrt{\frac{p}{n}} \tilde{K},$$

with

$$\tilde{K} = \left\{ \sum_{i,k=1}^{p} (\Gamma_i^2 + \sqrt{3(1 + 2\Gamma_i^2 + 3\Gamma_i^4)}\Gamma_{ik}^2 + 2\Gamma_i^2\Gamma_{ik})\|\alpha + \beta|Y_i|^\gamma\|_{L_4}^2 \|\alpha + \beta|Y_k|^\gamma\|_{L_4}^2 \right\}^{1/2}$$

$$= \left\{ \sum_{i,k=1}^{p} (\Gamma_i^2 + \sqrt{3(1 + 2\Gamma_i^2 + 3\Gamma_i^4)}\Gamma_{ik}^2 + 2\Gamma_i^2\Gamma_{ik})\|\alpha + \beta|\Gamma_i Z|^\gamma\|_{L_4}^2 \|\alpha + \beta|\Gamma_k Z|^\gamma\|_{L_4}^2 \right\}^{1/2},$$

with $Z \sim \mathcal{N}(0, 1)$, which concludes the proof.

# E   Gradient and Hessian for the output of a deep NN

The first step of the proofs of Theorem 4.2 and 4.3 is computing the gradient and the Hessian of $\tilde{F}$.

### E.1   $L = 2$

If $L = 2$, then

$$\tilde{F} = \sigma_w n^{-1/2} \sum_{i=1}^{n} w_i \tau(f_i^{(2)}(\boldsymbol{x}, n)),$$

where

$$f_i^{(2)}(\boldsymbol{x}, n) = \sigma_w n^{-1/2} \sum_{j=1}^n w_{i,j}^{(1)} \tau(f_j^{(1)}(\boldsymbol{x})),$$

$$f_j^{(1)}(\boldsymbol{x}) = \Gamma Y_j,$$

with $\Gamma^2 = \sigma_w^2 \|\boldsymbol{x}\|_2^2$. The partial derivatives are given by

$$
\begin{cases}
\dfrac{\partial \tilde{F}}{\partial w_i} = \sigma_w n^{-1/2} \tau\left(f_i^{(2)}(\boldsymbol{x}, n)\right) \\[3mm]
\dfrac{\partial \tilde{F}}{\partial w_{i,j}^{(1)}} = \left(\sigma_w n^{-1/2}\right)^2 w_i \tau\left(f_j^{(1)}(\boldsymbol{x})\right) \tau'\left(f_i^{(2)}(\boldsymbol{x}, n)\right) \\[3mm]
\dfrac{\partial \tilde{F}}{\partial Y_j} = \Gamma \left(\sigma_w n^{-1/2}\right)^2 \tau'\left(f_j^{(1)}(\boldsymbol{x})\right) \sum_{a=1}^n w_a w_{a,j}^{(1)} \tau'\left(f_a^{(2)}(\boldsymbol{x}, n)\right)
\end{cases}
$$

$$
\begin{cases}
\nabla^2_{w_i, w_j}\tilde{F} = 0 \\[3mm]
\nabla^2_{w_i, w_{k,j}^{(1)}}\tilde{F} = \delta_{ik}\left(\sigma_w n^{-1/2}\right)^2 \tau\left(f_j^{(1)}(\boldsymbol{x})\right)\tau'\left(f_i^{(2)}(\boldsymbol{x}, n)\right) \\[3mm]
\nabla^2_{w_i, Y_j}\tilde{F} = \Gamma\left(\sigma_w n^{-1/2}\right)^2 \tau'\left(f_j^{(1)}(\boldsymbol{x})\right)\tau'\left(f_i^{(2)}(\boldsymbol{x}, n)\right) \\[3mm]
\nabla^2_{w_{i,j}^{(1)}, w_{k,h}^{(1)}}\tilde{F} = \delta_{ik}\left(\sigma_w n^{-1/2}\right)^3 w_i \tau\left(f_j^{(1)}(\boldsymbol{x})\right)\tau\left(f_h^{(1)}(\boldsymbol{x})\right)\tau''\left(f_i^{(2)}(\boldsymbol{x}, n)\right) \\[3mm]
\nabla^2_{w_{i,j}^{(1)}, Y_k}\tilde{F} = \Gamma\left(\sigma_w n^{-1/2}\right)^2 w_i \tau'\left(f_k^{(1)}(\boldsymbol{x})\right)\left[\Gamma \sigma_w n^{-1/2} w_{i,k}^{(1)} \tau\left(f_j^{(1)}(\boldsymbol{x})\right)\tau''\left(f_i^{(2)}(\boldsymbol{x}, n)\right) + \delta_{jk}\tau'\left(f_i^{(2)}(\boldsymbol{x}, n)\right)\right] \\[3mm]
\nabla^2_{Y_j, Y_k}\tilde{F} = \left(\Gamma \sigma_w n^{-1/2}\right)^2 \Big[\sigma_w n^{-1/2}\tau'\left(f_j^{(1)}(\boldsymbol{x})\right)\tau'\left(f_k^{(1)}(\boldsymbol{x})\right)\sum_{a=1}^n w_a w_{a,j}^{(1)} w_{a,k}^{(1)} \tau''\left(f_a^{(2)}(\boldsymbol{x}, n)\right) \; + \\
\qquad\qquad + \delta_{jk}\tau''\left(f_j^{(1)}(\boldsymbol{x})\right)\sum_{a=1}^n w_a w_{a,j}^{(1)} \tau'\left(f_a^{(2)}(\boldsymbol{x}, n)\right)\Big],
\end{cases}
$$

and this for all $i, j, k \in [n]$.

## E.2 General $L$

In this section will compute the gradient and the hessian of the NN defined in (1) for a general $L$, not necessarily $L = 2$ as in the previous one. Application of Theorem 4.1 requires computing the gradient and the hessian of $F_i = f^{(L+1)}(\boldsymbol{x}_i)$, and it will be sufficient to use all the computations of this section with $F_i$ in place of $F$, and $\boldsymbol{x}_i$ in place of $\boldsymbol{x}$. To simplify the notation we write $f_i^{(l)}(\boldsymbol{x}) := f_i^{(l)}(\boldsymbol{x}, n)$ for every $i$ and $l$.

It is useful to start by computing the following derivatives

$$\frac{\partial F}{\partial f_{i_L}^{(L)}(\boldsymbol{x})} = \frac{\sigma_w}{\sqrt{n}} w_{i_L} \tau'\left(f_{i_L}^{(L)}(\boldsymbol{x})\right)$$

$$\frac{\partial f_{i_{l+1}}^{(l+1)}(\boldsymbol{x})}{\partial f_{i_l}^{(l)}(\boldsymbol{x})} = \frac{\sigma_w}{\sqrt{n}} w_{i_{l+1}, i_l}^{(l)} \tau'\left(f_{i_l}^{(l)}(\boldsymbol{x})\right) \quad \forall l \in \{1, \ldots, L-1\}$$

$$\frac{\partial f_{i_{l+1}}^{(l+1)}(\boldsymbol{x})}{\partial w_{i_l,j_l}^{(l)}} = \delta_{i_{l+1}i_l}\frac{\sigma_w}{\sqrt{n}}\tau\left(f_{j_l}^{(l)}(\boldsymbol{x})\right) \quad \forall l \in \{1,\dots,L-1\}$$

$$\frac{\partial f_{i_1}^{(1)}(\boldsymbol{x})}{\partial w_{i_0,j_0}^{(0)}} = \delta_{i_1 i_0}\sigma_w x_{j_0},$$

which hold true for all $i_L,\dots,i_0,j_L,\dots,j_1 = 1,\dots,n$ and $j_0 = 1,\dots,d$.
Using the chain rule, it is easy but a little tedious to compute

$$\frac{\partial F}{\partial w_{i_L}} = \frac{\sigma_w}{\sqrt{n}}\tau\left(f_{i_L}^{(L)}(\boldsymbol{x})\right)$$

$$\frac{\partial F}{\partial w_{i_{L-1},j_{L-1}}^{(L-1)}} = \left(\frac{\sigma_w}{\sqrt{n}}\right)^2 w_{i_{L-1}}\tau'\left(f_{i_{L-1}}^{(L)}(\boldsymbol{x})\right)\tau\left(f_{j_{L-1}}^{(L-1)}(\boldsymbol{x})\right)$$

$$\frac{\partial F}{\partial w_{i_{L-2},j_{L-2}}^{(L-2)}} = \left(\frac{\sigma_w}{\sqrt{n}}\right)^3 \tau'\left(f_{i_{L-2}}^{(L-1)}(\boldsymbol{x})\right)\tau\left(f_{j_{L-2}}^{(L-2)}(\boldsymbol{x})\right)\sum_{i_L=1}^n w_{i_L}\tau'\left(f_{i_L}^{(L)}(\boldsymbol{x})\right)w_{i_L,i_{L-2}}^{(L-1)}$$

$$\frac{\partial F}{\partial w_{i_l,j_l}^{(l)}} = \left(\frac{\sigma_w}{\sqrt{n}}\right)^{L-l+1}\tau'\left(f_{i_l}^{(l+1)}(\boldsymbol{x})\right)\tau\left(f_{j_l}^{(l)}(\boldsymbol{x})\right)\times$$

$$\times \sum_{i_L,\dots,i_{l+2}=1}^n w_{i_L}\tau'\left(f_{i_L}^{(L)}(\boldsymbol{x})\right)\left(\prod_{s=l+2}^{L-1}w_{i_{s+1},i_s}^{(s)}\tau'\left(f_{i_s}^{(s)}(\boldsymbol{x})\right)\right)w_{i_{l+2},i_l}^{(l+1)}$$

$$\frac{\partial F}{\partial w_{i_0,j_0}^{(0)}} = \sigma_w\left(\frac{\sigma_w}{\sqrt{n}}\right)^L\tau'\left(f_{i_0}^{(1)}(\boldsymbol{x})\right)x_{j_0}\times$$

$$\times \sum_{i_L,\dots,i_2=1}^n w_{i_L}\tau'\left(f_{i_L}^{(L)}(\boldsymbol{x})\right)\left(\prod_{s=2}^{L-1}w_{i_{s+1},i_s}^{(s)}\tau'\left(f_{i_s}^{(s)}(\boldsymbol{x})\right)\right)w_{i_2,i_0}^{(1)}$$

for all $i_L,\dots,i_0,j_L,\dots,j_1 = 1,\dots,n$, $j_0 = 1,\dots,d$ and $l = 1,\dots,L-3$.

As for the Hessian, we have

$$\nabla^2_{w_{i_L},w_{j_L}}F = 0$$

$$\nabla^2_{w_{i_L},w_{i_{L-1},j_{L-1}}^{(L-1)}}F = \delta_{i_L i_{L-1}}\left(\frac{\sigma_w}{\sqrt{n}}\right)^2\tau'\left(f_{i_L}^{(L)}(\boldsymbol{x})\right)\tau\left(f_{j_{L-1}}^{(L-1)}(\boldsymbol{x})\right)$$

$$\nabla^2_{w_{i_L},w_{i_{L-2},j_{L-2}}^{(L-2)}}F = \left(\frac{\sigma_w}{\sqrt{n}}\right)^3\tau'\left(f_{i_L}^{(L)}(\boldsymbol{x})\right)\tau'\left(f_{i_{L-2}}^{(L-1)}(\boldsymbol{x})\right)\tau\left(f_{j_{L-2}}^{(L-2)}(\boldsymbol{x})\right)w_{i_L,i_{L-2}}^{(L-1)}$$

$$\nabla^2_{w_{i_L},w_{i_l,j_l}^{(l)}}F = \left(\frac{\sigma_w}{\sqrt{n}}\right)^{L-l+1}\tau'\left(f_{i_L}^{(L)}(\boldsymbol{x})\right)\tau'\left(f_{i_l}^{(l+1)}(\boldsymbol{x})\right)\tau\left(f_{j_l}^{(l)}(\boldsymbol{x})\right)\times$$

$$\times \sum_{i_{L-1},\dots,i_{l+2}=1}^n\left(\prod_{s=l+2}^{L-1}w_{i_{s+1},i_s}^{(s)}\tau'\left(f_{i_s}^{(s)}(\boldsymbol{x})\right)\right)w_{i_{l+2},i_l}^{(l+1)}$$

$$\nabla^2_{w_{i_L},w_{i_0,j_0}^{(0)}}F = \sigma_w\left(\frac{\sigma_w}{\sqrt{n}}\right)^L\tau'\left(f_{i_L}^{(L)}(\boldsymbol{x})\right)\tau'\left(f_{i_0}^{(1)}(\boldsymbol{x})\right)x_{j_0}\times$$

$$\times \sum_{i_{L-1},\dots,i_2=1}^n\left(\prod_{s=2}^{L-1}w_{i_{s+1},i_s}^{(s)}\tau'\left(f_{i_s}^{(s)}(\boldsymbol{x})\right)\right)w_{i_2,i_0}^{(1)}$$

As for two generic weights $w_{i_l,j_l}^{(l)}, w_{j_m,\tilde{j}_m}^{(m)}$ for $l \in \{0, \dots, L-1\}$, we have

$$
\nabla^2_{w_{i_l,j_l}^{(l)}, w_{j_m,\tilde{j}_m}^{(m)}} F = \left(\frac{\sigma_w}{\sqrt{n}}\right)^{L-l+1} \frac{\partial}{\partial w_{j_m,\tilde{j}_m}^{(m)}} \left[\tau'\left(f_{i_l}^{(l+1)}(\boldsymbol{x})\right)\right] \tau\left(f_{j_l}^{(l)}(\boldsymbol{x})\right) \times
$$

$$
\times \sum_{i_L,\dots,i_{l+2}=1}^n w_{i_L} \tau'\left(f_{i_L}^{(L)}(\boldsymbol{x})\right) \left(\prod_{s=l+2}^{L-1} w_{i_{s+1},i_s}^{(s)} \tau'\left(f_{i_s}^{(s)}(\boldsymbol{x})\right)\right) w_{i_{l+2},i_l}^{(l+1)} +
$$

$$
+ \left(\frac{\sigma_w}{\sqrt{n}}\right)^{L-l+1} \tau'\left(f_{i_l}^{(l+1)}(\boldsymbol{x})\right) \frac{\partial}{\partial w_{j_m,\tilde{j}_m}^{(m)}} \left[\tau\left(f_{j_l}^{(l)}(\boldsymbol{x})\right)\right] \times
$$

$$
\times \sum_{i_L,\dots,i_{l+2}=1}^n w_{i_L} \tau'\left(f_{i_L}^{(L)}(\boldsymbol{x})\right) \left(\prod_{s=l+2}^{L-1} w_{i_{s+1},i_s}^{(s)} \tau'\left(f_{i_s}^{(s)}(\boldsymbol{x})\right)\right) w_{i_{l+2},i_l}^{(l+1)} +
$$

$$
+ \left(\frac{\sigma_w}{\sqrt{n}}\right)^{L-l+1} \tau'\left(f_{i_l}^{(l+1)}(\boldsymbol{x})\right) \tau\left(f_{j_l}^{(l)}(\boldsymbol{x})\right) \times
$$

$$
\times \left[\sum_{i_L,\dots,i_{l+2}=1}^n \beta_{i_L,\dots,i_{l+2}} \frac{\partial}{\partial w_{j_m,\tilde{j}_m}^{(m)}} \alpha_{i_L,\dots,i_{l+2}} + \beta_{i_L,\dots,i_{l+2}} \frac{\partial}{\partial w_{j_m,\tilde{j}_m}^{(m)}} \alpha_{i_L,\dots,i_{l+2}}\right]
$$

where

$$
\alpha_{i_L,\dots,i_{l+2}} := \left(w_{i_L} w_{i_{l+2},i_l}^{(l+1)} \prod_{s=l+2}^{L-1} w_{i_{s+1},i_s}^{(s)}\right)
$$

and

$$
\beta_{i_L,\dots,i_{l+2}} := \left(\tau'\left(f_{i_L}^{(L)}(\boldsymbol{x})\right) \prod_{s=l+2}^{L-1} \tau'\left(f_{i_s}^{(s)}(\boldsymbol{x})\right)\right),
$$

so that

$$
\frac{\partial}{\partial w_{j_m,\tilde{j}_m}^{(m)}} \alpha_{i_L,\dots,i_{l+2}} = \delta_{j_m,i_{m+1}} \delta_{\tilde{j}_m,i_m} \mathbb{1}\{m > l+1\} \left(\frac{w_{i_L} w_{i_{l+2},i_l}^{(l+1)}}{w_{j_m,\tilde{j}_m}^{(m)}} \prod_{s=l+2}^{L-1} w_{i_{s+1},i_s}^{(s)}\right)
$$

$$
+ \delta_{j_m,i_{l+2}} \delta_{\tilde{j}_m,i_l} \mathbb{1}\{m = l+1\} \left(w_{i_L} \prod_{s=l+2}^{L-1} w_{i_{s+1},i_s}^{(s)}\right),
$$

$$
\frac{\partial}{\partial w_{j_m,\tilde{j}_m}^{(m)}} \beta_{i_L,\dots,i_{l+2}} = \beta_{i_L,\dots,i_{l+2}} \sum_{s=l+2}^L \frac{1}{\tau'\left(f_{i_s}^{(s)}(\boldsymbol{x})\right)} \frac{\partial}{\partial w_{j_m,\tilde{j}_m}^{(m)}} \tau'\left(f_{i_s}^{(s)}(\boldsymbol{x})\right),
$$

with

$$
\frac{\partial}{\partial w_{i_m,j_m}^{(m)}} \left[\tau'\left(f_{i_l}^{(l+1)}(\boldsymbol{x})\right)\right] = \begin{cases} 0 & \text{if } m \geq l+1 \\[2ex] \frac{\sigma_w}{\sqrt{n}} \delta_{i_l i_m} \tau''\left(f_{i_l}^{(l+1)}(\boldsymbol{x})\right) \tau\left(f_{j_m}^{(m)}(\boldsymbol{x})\right) & \text{if } m = l \\[2ex] \left(\frac{\sigma_w}{\sqrt{n}}\right)^2 \tau''\left(f_{i_l}^{(l+1)}(\boldsymbol{x})\right) \tau'\left(f_{i_m}^{(m+1)}(\boldsymbol{x})\right) \tau\left(f_{j_m}^{(m)}(\boldsymbol{x})\right) w_{i_l,i_m}^{(m+1)} & \text{if } m = l-1 \\[2ex] \left(\frac{\sigma_w}{\sqrt{n}}\right)^{l-m+1} \tau''\left(f_{i_l}^{(l+1)}(\boldsymbol{x})\right) \tau'\left(f_{i_m}^{(m+1)}(\boldsymbol{x})\right) \tau\left(f_{j_m}^{(m)}(\boldsymbol{x})\right) \times \\ \quad \times \sum_{k_l,\dots,k_{m+2}=1}^n w_{i_l,k_l}^{(l)} \tau'\left(f_{k_l}^{(l)}(\boldsymbol{x})\right) \left(\prod_{s=m+2}^{l-1} w_{k_{s+1},k_s}^{(s)} \tau'\left(f_{k_s}^{(s)}(\boldsymbol{x})\right)\right) w_{k_{m+2},i_m}^{(m+1)} & \text{if } m < l-1 \end{cases}
$$

and

$$
\frac{\partial}{\partial w_{i_m,j_m}^{(m)}} \left[ \tau\left(f_{j_l}^{(l)}(\boldsymbol{x})\right)\right] = \begin{cases} 0 & \text{if } m \geq l \\[2mm] \frac{\sigma_w}{\sqrt{n}} \delta_{j_l i_m} \tau'\left(f_{j_l}^{(l)}(\boldsymbol{x})\right) \tau\left(f_{j_m}^{(m)}(\boldsymbol{x})\right) & \text{if } m = l-1 \\[3mm] \left(\frac{\sigma_w}{\sqrt{n}}\right)^2 \tau'\left(f_{j_l}^{(l)}(\boldsymbol{x})\right) \tau'\left(f_{i_m}^{(m+1)}(\boldsymbol{x})\right) \tau\left(f_{j_m}^{(m)}(\boldsymbol{x})\right) w_{j_l,i_m}^{(m+1)} & \text{if } m = l-2 \\[3mm] \left(\frac{\sigma_w}{\sqrt{n}}\right)^{l-m} \tau'\left(f_{j_l}^{(l)}(\boldsymbol{x})\right) \tau'\left(f_{i_m}^{(m+1)}(\boldsymbol{x})\right) \tau\left(f_{j_m}^{(m)}(\boldsymbol{x})\right) \times \\[2mm] \quad \times \sum_{k_{l-1},\dots,k_{m+2}=1}^{n} w_{j_l,k_{l-1}}^{(l-1)} \tau'\left(f_{k_{l-1}}^{(l)}(\boldsymbol{x})\right) \left(\prod_{s=m+2}^{l-2} w_{k_{s+1},k_s}^{(s)} \tau'\left(f_{k_s}^{(s)}(\boldsymbol{x})\right)\right) w_{k_{m+2},i_m}^{(m+1)} & \text{if } m < l-2. \end{cases}
$$

## F   Proof of Theorems 4.2 and Theorem 4.3

We will write $a \lesssim b$ if there exists a universal constant $C$ such that $a \leq Cb$, and $a \asymp b$ if both $a \lesssim b$ and $b \lesssim a$. Both the proofs are essentially based on Theorem 4.1, adapted to the case $L = 2$, and $p = 1$ and $p \geq 2$ respectively. As outlined in the main body, after stating Theorem 4.1, the biggest problem one has to face lies in the fact that there is not a straightforward way of controlling the expectations in the bound, since each node depends on the nodes of all the previous layers in a very convoluted manner. Nonetheless, it is still possible to overcome this problem in this case by conditioning on the previous hidden layer, since $f_{\cdot}^{(1)}(\boldsymbol{x})$ is normally distributed. We will show how to do this for a specific term in the bound, as for the others the same methodology can be applied. To simplify the notation, we will write $f_i^{(2)}(\boldsymbol{x}) := f_i^{(2)}(\boldsymbol{x}, n)$.

Without loss of generality, we can assume $\gamma > 1$.

We will make use several times of the following generalized Bahr-Esseen inequalities (Dharmadhikari & Jogdeo, 1969): if $X_1, \dots, X_n$ are independent, zero mean random variables with finite $r$-th moment, for some $r > 2$, then

$$
\mathbb{E}\left[\left|\sum_{k=1}^{n} X_k\right|^r\right] \leq c n^{r/2-1} \sum_{k=1}^{n} \mathbb{E}[|X_k|^r]
$$

where $c > 0$ is a constant that depends only on $r$.

First, notice that, for every $r > 2$, $\mathbb{E}[|f_i^{(1)}(\boldsymbol{x})|^r]$ is bounded by a constant that only depends on $r$ and $\boldsymbol{x}$. Moreover, for every $r > 0$,

$$
\begin{aligned}
\mathbb{E}\left[|\tau(f_i^{(2)}(\boldsymbol{x})|^r \mid Y_{\cdot}\right] &\leq \mathbb{E}\left[(\alpha + \beta|f_i^{(2)}(\boldsymbol{x})|)^r \mid Y_{\cdot}\right] \\
&\leq 2^r \left(\alpha^r + \beta^r \mathbb{E}[|f_i^{(2)}(\boldsymbol{x})|^r \mid Y_{\cdot}]\right) \\
&\leq 2^r \alpha^r + 2^r \beta^r \sigma_w^r n^{-r/2} \mathbb{E}\left[|\sum_{j=1}^{n} w_{i,j}^{(1)} \tau(f_j^{(1)}(\boldsymbol{x}))|^r \mid Y_{\cdot}\right] \\
&\leq 2^r \alpha^r + 2^r \beta^r \sigma_w^r n^{-1} \sum_{j=1}^{n} \mathbb{E}\left[|w_{i,j}^{(1)} \tau(f_j^{(1)}(\boldsymbol{x}))|^r \mid Y_{\cdot}\right] \\
&\leq 2^r \alpha^r + 2^r \beta^r \sigma_w^r \mathbb{E}[|Z|^r] n^{-1} \sum_{j=1}^{n} |\tau(f_j^{(1)}(\boldsymbol{x}))|^r,
\end{aligned}
$$

where $Z \sim \mathcal{N}(0, 1)$ and we have used the generalized Bahr-Esseen inequality and the fact that the random variables $w_{i,j}^{(1)} \tau(f_j^{(1)}(\boldsymbol{x}))$ are conditionally independent, given $Y_{\cdot}$, with zero conditional expectations. The same equations apply to $|\tau'(f_i^{(2)}(\boldsymbol{x}))|$. It follows that, for every $r > 0$ there exists a $c_r$ not depending on $n$ such that

$$
\max\left(\mathbb{E}\left[|\tau(f_i^{(2)}(\boldsymbol{x})|^r\right], \mathbb{E}\left[|\tau'(f_i^{(2)}(\boldsymbol{x})|^r\right]\right) \leq c_r.
$$

We will now show how to bound $\sum_{i,j=1}^{n} \left\{ \mathbb{E}\left[ \left( \frac{\partial F}{\partial w_i} \frac{\partial F}{\partial w_j} \right)^2 \right] \mathbb{E}\left[ \left\langle \nabla_{w_i,\cdot}^2 F, \nabla_{w_j,\cdot}^2 F \right\rangle^2 \right] \right\}^{1/2}$ from above. If $i \neq j$, then

$$\mathbb{E}\left[ \left( \frac{\partial F}{\partial w_i} \frac{\partial F}{\partial w_j} \right)^2 \right] = (\sigma_w n^{-1/2})^4 \mathbb{E}\left[ \mathbb{E}\left[ \tau^2(f_i^{(2)}(x))|Y_\cdot \right]^2 \right] \leq \sigma_w^4 n^{-2} \mathbb{E}\left[ \tau^4(f_i^{(2)}(x)) \right] \lesssim n^{-2}.$$

For $i = j$, we can write that

$$\mathbb{E}\left[ \left( \frac{\partial F}{\partial w_i} \right)^4 \right] \leq \sigma_w^4 n^{-2} \mathbb{E}\left[ \tau^4(f_i^{(2)}(x)) \right] \lesssim n^{-2}.$$

Let us now turn to the expectation involving the Hessian. We have that

$$\left\langle \nabla_{w_i,\cdot}^2 F, \nabla_{w_j,\cdot}^2 F \right\rangle = \left( \frac{\sigma_w}{\sqrt{n}} \right)^4 \tau'\left( f_i^{(2)}(\boldsymbol{x}) \right) \tau'\left( f_j^{(2)}(\boldsymbol{x}) \right) \left( \delta_{ij} \sum_{b=1}^{n} \tau\left( f_b^{(1)}(\boldsymbol{x}) \right)^2 + \Gamma^2 \sum_{b=1}^{n} w_{i,b}^{(1)} w_{j,b}^{(1)} \tau'\left( f_b^{(1)}(\boldsymbol{x}) \right)^2 \right),$$

so that

$$\left\langle \nabla_{w_i,\cdot}^2 F, \nabla_{w_j,\cdot}^2 F \right\rangle^2 \leq \left( \frac{\sigma_w}{\sqrt{n}} \right)^8 \tau'\left( f_i^{(2)}(\boldsymbol{x}) \right)^2 \tau'\left( f_j^{(2)}(\boldsymbol{x}) \right)^2 \times$$

$$\times \left( 2\delta_{ij} \left( \sum_{b=1}^{n} \tau\left( f_b^{(1)}(\boldsymbol{x}) \right)^2 \right)^2 + 2\Gamma^4 \left( \sum_{b=1}^{n} w_{i,b}^{(1)} w_{j,b}^{(1)} \tau'\left( f_b^{(1)}(\boldsymbol{x}) \right)^2 \right)^2 \right)$$

$$\lesssim n^{-4} \delta_{ij} \tau'\left( f_i^{(2)}(\boldsymbol{x}) \right)^2 \tau'\left( f_j^{(2)}(\boldsymbol{x}) \right)^2 \left( \sum_{b=1}^{n} \tau\left( f_b^{(1)}(\boldsymbol{x}) \right)^2 \right)^2$$

$$+ n^{-4} \tau'\left( f_i^{(2)}(\boldsymbol{x}) \right)^2 \tau'\left( f_j^{(2)}(\boldsymbol{x}) \right)^2 \left( \sum_{b=1}^{n} w_{i,b}^{(1)} w_{j,b}^{(1)} \tau'\left( f_b^{(1)}(\boldsymbol{x}) \right)^2 \right)^2.$$

We will bound the expectations of the two terms of the sum separately. For the first term we have

$$\mathbb{E}\left[ n^{-4} \delta_{ij} \tau'\left( f_i^{(2)}(\boldsymbol{x}) \right)^2 \tau'\left( f_j^{(2)}(\boldsymbol{x}) \right)^2 \left( \sum_{b=1}^{n} \tau\left( f_b^{(1)}(\boldsymbol{x}) \right)^2 \right)^2 \right]$$

$$\leq n^{-4} \delta_{ij} \mathbb{E}\left[ \left( \sum_{b=1}^{n} \tau\left( f_b^{(1)}(\boldsymbol{x}) \right)^2 \right)^2 \mathbb{E}\left[ \tau'\left( f_i^{(2)}(\boldsymbol{x}) \right)^2 |Y_\cdot \right]^2 \right]$$

$$\leq n^{-4} \delta_{ij} \mathbb{E}\left[ \left( \sum_{b=1}^{n} \tau\left( f_b^{(1)}(\boldsymbol{x}) \right)^2 \right)^2 \mathbb{E}\left[ \tau'\left( f_i^{(2)}(\boldsymbol{x}) \right)^4 |Y_\cdot \right] \right]$$

$$\leq n^{-4} \delta_{ij} \left( \mathbb{E}\left[ \left( \sum_{b=1}^{n} \tau\left( f_b^{(1)}(\boldsymbol{x}) \right)^2 \right)^4 \right] \right)^{1/2} \left( \mathbb{E}\left[ \tau'\left( f_i^{(2)}(\boldsymbol{x}) \right)^8 \right] \right)^{1/2}$$

$$\lesssim n^{-2} \delta_{ij},$$

For the second term, we consider the cases $i = j$ and $i \neq j$ separately. For $i = j$ we can write that

$$\mathbb{E}\left[ n^{-4} \delta_{ij} \tau'\left( f_i^{(2)}(\boldsymbol{x}) \right)^4 \left( \sum_{b=1}^{n} (w_{i,b}^{(1)})^2 \tau'\left( f_b^{(1)}(\boldsymbol{x}) \right)^2 \right)^2 \right]$$

$$\leq n^{-4} \delta_{ij} c_8^{1/2} \left( \mathbb{E}\left[ \left( \sum_{b=1}^{n} (w_{i,b}^{(1)})^2 \tau'\left( f_b^{(1)}(\boldsymbol{x}) \right)^2 \right)^4 \right] \right)^{1/2}$$

$$\lesssim \delta_{ij} n^{-2}.$$

On the other hand, for $i \neq j$ we can write that

$$\mathbb{E}\left[ n^{-4} \tau'\left(f_i^{(2)}(\boldsymbol{x})\right)^2 \tau'\left(f_j^{(2)}(\boldsymbol{x})\right)^2 \left( \sum_{b=1}^{n} w_{i,b}^{(1)} w_{j,b}^{(1)} \tau'\left(f_b^{(1)}(\boldsymbol{x})\right)^2 \right)^2 \right]$$

$$\leq n^{-4} \left( \mathbb{E}\left[ \tau'\left(f_i^{(2)}(\boldsymbol{x})\right)^4 \tau'\left(f_j^{(2)}(\boldsymbol{x})\right)^4 \right] \right)^{1/2} \left( \mathbb{E}\left[ \left( \sum_{b=1}^{n} w_{i,b}^{(1)} w_{j,b}^{(1)} \tau'\left(f_b^{(1)}(\boldsymbol{x})\right)^2 \right)^4 \right] \right)^{1/2}$$

$$\leq n^{-4} c_8^{1/2} \left( \mathbb{E}\left[ \mathbb{E}\left[ \left( \sum_{b=1}^{n} w_{i,b}^{(1)} w_{j,b}^{(1)} \tau'\left(f_b^{(1)}(\boldsymbol{x})\right)^2 \right)^4 \mid Y. \right] \right] \right)^{1/2}$$

$$\leq n^{-4} c_8^{1/2} \left( \mathbb{E}\left[ n \sum_{b=1}^{n} \mathbb{E}\left[ |w_{i,b}^{(1)} w_{j,b}^{(1)}|^4 |\tau'\left(f_b^{(1)}(\boldsymbol{x})\right)|^8 \mid Y. \right] \right] \right)^{1/2}$$

$$\leq n^{-4} c_8^{1/2} \left( \mathbb{E}\left[ n \sum_{b=1}^{n} |\tau'\left(f_b^{(1)}(\boldsymbol{x})\right)|^8 \mathbb{E}\left[ |w_{i,b}^{(1)} w_{j,b}^{(1)}|^4 \mid Y. \right] \right] \right)^{1/2}$$

$$\leq n^{-4} c_8^{1/2} E\left[ |Z|^4 \right] \left( n \mathbb{E}\left[ \sum_{b=1}^{n} |\tau'\left(f_b^{(1)}(\boldsymbol{x})\right)|^8 \right] \right)^{1/2}$$

$$\lesssim n^{-3}$$

Summarizing, we can write that

$$\sum_{i,j=1}^{n} \left\{ \mathbb{E}\left[ \left( \frac{\partial F}{\partial w_i} \frac{\partial F}{\partial w_j} \right)^2 \right] \mathbb{E}\left[ \left\langle \nabla^2_{w_i, \cdot} F, \nabla^2_{w_j, \cdot} F \right\rangle^2 \right] \right\}^{1/2} \lesssim \sum_{i,j=1}^{n} \{ n^{-2} (\delta_{ij} n^{-2} + n^{-3}) \}^{1/2} \lesssim n^{-1/2}.$$

The same rate can be found with analogous steps for all the other terms in the sum given by Theorem 4.1, and taking the square root one more time gives the rate of $n^{-1/4}$. The proof in the case of $p$ output is essentially the same, apart from the fact that we have an extra sum over $p$ index, which leads to the rate $\mathcal{O}(\sqrt{p/\sqrt{n}})$.

As stated in the main body, this rate is worse than the one in Basteri & Trevisan (2022), but in order to show that this in not "our fault", but it is due to the intrinsic behaviour of these Poincaré inequality in this setting, we will now show that the same rate $n^{-1/4}$ is obtained in the case $\tau = id$, the identity function, which is arguably the nicest setting possible. Indeed, if we consider the NN

$$F := n^{-1} \sum_{i=1}^{n} \sum_{j=1}^{n} w_i w_{i,j}^{(1)} Y_j,$$

we can compute explicitly

$$\mathbb{E}\left[ \left( \frac{\partial F}{\partial w_i} \frac{\partial F}{\partial w_j} \right)^2 \right] \quad \text{and} \quad \mathbb{E}\left[ \left\langle \nabla^2_{w_i, \cdot} F, \nabla^2_{w_j, \cdot} F \right\rangle^2 \right],$$

and see that they lead to the same suboptimal rate of $n^{-1/4}$. As for the first term, we have

$$\mathbb{E}\left[ \left( \frac{\partial F}{\partial w_i} \frac{\partial F}{\partial w_j} \right)^2 \right] = n^{-2} \mathbb{E}\left[ \mathbb{E}\left[ (f_i^{(2)}(x))^2 | Y. \right]^2 \right],$$

and

$$\mathbb{E}\left[(f_i^{(2)}(x))^2|Y_.\right] = n^{-1}\mathbb{E}\left[\left(\sum_{j=1}^n w_{i,j}^{(1)}Y_j\right)^2\Big|Y_.\right] = n^{-1}\mathbb{E}\left[\sum_{j,k=1}^n w_{i,j}^{(1)}w_{i,k}^{(1)}Y_jY_k\Big|Y_.\right] = n^{-1}\sum_{j=1}^n Y_j^2,$$

so that

$$\mathbb{E}\left[\left(\frac{\partial F}{\partial w_i}\frac{\partial F}{\partial w_j}\right)^2\right] = n^{-4}\mathbb{E}\left[\left(\sum_{j=1}^n Y_j^2\right)^2\right] = n^{-4}\sum_{j,k=1}^n \mathbb{E}\left[Y_j^2 Y_k^2\right] \asymp n^{-2}.$$

As for the second term,

$$\mathbb{E}\left[\left\langle\nabla_{w_i,.}^2 F, \nabla_{w_j,.}^2 F\right\rangle\right] \asymp n^{-4}\delta_{ij}\mathbb{E}\left[\left(\sum_{b=1}^n Y_b^2\right)^2\right] + n^{-4}\mathbb{E}\left[\left(\sum_{b=1}^n w_{i,b}^{(1)}w_{j,b}^{(1)}\right)^2\right]$$

$$= n^{-4}\delta_{ij}(2n+n^2) + n^{-4}\mathbb{E}\left[\left(\sum_{b=1}^n w_{i,b}^{(1)}w_{j,b}^{(1)}\right)^2\right],$$

since $\sum_{b=1}^n Y_b^2 \sim \chi_n^2$, and $\mathbb{E}[\chi_n^2] = n$ and $\mathrm{Var}[\chi_n^2] = 2n$. Also,

$$\mathbb{E}\left[\left(\sum_{b=1}^n w_{i,b}^{(1)}w_{j,b}^{(1)}\right)^2\right] = \mathbb{E}\left[\sum_{a,b=1}^n w_{i,b}^{(1)}w_{j,b}^{(1)}w_{i,a}^{(1)}w_{j,a}^{(1)}\right] = \sum_{a,b=1}^n \mathbb{E}\left[w_{i,b}^{(1)}w_{j,b}^{(1)}w_{i,a}^{(1)}w_{j,a}^{(1)}\right]$$

$$\lesssim \sum_{a,b=1}^n \left[\delta_{ij} + (1-\delta_{ij})\delta_{ab}\right] = n^2\delta_{ij} + n(1-\delta_{ij}),$$

hence

$$\mathbb{E}\left[\left\langle\nabla_{w_i,.}^2 F, \nabla_{w_j,.}^2 F\right\rangle^2\right] \lesssim n^{-4}\left[\delta_{ij}(2n+n^2) + n^2\delta_{ij} + n(1-\delta_{ij})\right] \lesssim n^{-2}\delta_{ij} + (1-\delta_{ij})n^{-3}.$$

Combining the two terms we get something of the order $n^{-4}\delta_{ij} + (1-\delta_{ij})n^{-5}$, and after taking the square root, something like $\sqrt{n^{-4}\delta_{ij} + (1-\delta_{ij})n^{-5}} \lesssim n^{-2}\delta_{ij} + (1-\delta_{ij})n^{-5/2}$. The same is true for all the others terms which appear in the bound of Theorem 4.1, hence, summing over all $i,j \in [n]$, gives a rate whose leading term is again of the order $n^{-1/4}$.

