# OpenReview forum: "Non-asymptotic approximations of Gaussian neural networks via second-order Poincar\'e inequalities"
_TMLR — Rejected by TMLR_

### Review · Reviewer_UiBT · 2023-08-25

**Summary Of Contributions:**

The authors attempt to match known rates for the GP-limiting behavior of wide neural networks, using an alternative proof technique. Their proof technique matches known upper bounds for 1-hidden layer networks, but is looser for 2-hidden layer networks. This looseness is conjectured to be a fundamental limitation of their proof technique.

**Audience:**

No

**Broader Impact Concerns:**

None.

**Claims And Evidence:**

No

**Requested Changes:**

## Claims and Evidence
In order to meet the "Claims and Evidence" criterion, the authors would need to make the paper more clear and readable by doing the following:

### Major
- Rewrite the proofs of 4.2 and 4.3 for additional clarity. Consider the first equation block: the explanation for why $E[|f_i(1)(\boldsymbol x)|^r]$ is bounded by a constant is not clear. Nor is it clear where the $2^r$ term comes from, nor where the $n^{-r/2}$ term in the third inequality, etc. This proof is much less clear than the others.

### Minor
- Proof of 3.2: "But since $\tau$ is polynomially bounded and the square root is an increasing function, we can bound this expression by..." - it would be nice to be a bit more explicit about the bound on $\tau$ leads to bounds on $\tau'$ and $\tau''$.
- Theorems 3.1 and 3.2 are nice, because they serve as "warm-ups" to Theorem 3.3. However, I don't think that they actually are a result that should be mentioned in the main text (no one actually uses the R.V. defined by 9, and Theorem 3.3 is a strict generalization of Theorem 3.2.)
- Calling Eq. (9) a NN is very strange, since it is not a function. You should refer to it as a random variable.
- In the intro and Section 4, the authors say "... leading to a worst [sic] rate of convergence than $n^{-1/2}$, which is expected from Basteri & Trevisan (2022), ..." This phrasing is confusing. I first interpreted this sentence to mean that the results of Basteri & Trevisan + others suggest a worse rate of convergence for the $L \geq 2$ case. Better phrasing would be "... leading to a worse rate of convergence than the $n^{-1/2}$ obtained by Basteri & Trevisan (2022), ..."

### Very minor
- $\Vert h \Vert_\mathrm{Lip}$ is strange notation for a Lipschitz constant, because (as far as I am aware) the Lipschitz constant is not a norm?

## Audience
I'm honestly not sure what changes I would expect to see that would address my "Audience" concerns (see above). I would be open to a rebuttal from the authors, as well as my fellow reviewers, for why they think that these results will have interest from the larger community.

**Strengths And Weaknesses:**

## Strengths
- The proof technique is nice for the $L=1$ case. As the authors point out, it matches known bounds on the CLT behavior of wide neural networks, while essentially only relying on algebraic computations of the Hessian and derivatives of NN.
- The paper is mostly clear, with the notable exception of Appendix F (see Weaknesses)

## Weaknesses (Claims and Evidence)
- The proof in Appendix F is very hard to follow, and as a result I was unable to verify its correctness. See requested changes.

## Weaknesses (Audience)
The acceptance criterion for TMLR is "would some individuals in TMLR's audience be interested in the findings of this paper?" I debated this question in my mind for a while, and currently my answer to this question is "no."

At a traditional ML venue (e.g. NeurIPS, JMLR, ICLR, etc.), there would be significant concerns about the significance/impact/novelty of this paper. Of course, TMLR explicitly avoids this language in its criteria for accepted papers, but even still it is challenging to think of why this paper would be of interest. To a large extent, it is a negative result - showing that an alternative (and arguably well-established proof technique) cannot match the bounds of current work. Negative results can be interesting, but the authors have not made the case for why this negative result would be interesting. To make this result more interesting, the authors would have had to answer the following:

- What was the original purpose for investigating Poincare inequalities in the first place? A simpler proof? Matching Wasserstein-2 rates for different distance metrics? Hope for tighter bounds? (The latter seems unlikely, given that $n^{-1/2}$ rate already obtained in a better setting.)
- What insight can others gain from the negative result? In my mind, it should be a warning that Poincare inequalities are not a fruitful direction for limiting behavior of deep models (given that the $n^{-1/4}$ rate seems to be a limitation of the proof technique). This takeaway would be in direct contradiction to the authors' claims in "the effectiveness of second-order Poincare inequalities... in establishing an quantitative CLT for a complicated functional of Gaussian processes such as the deep Gaussian NN."

If the authors do want to meet the interest criterion, they need to make a proper case for why this negative result should be of interest.

---

### Review · Reviewer_NyyW · 2023-09-06

**Summary Of Contributions:**

The authors study quantitative rates of convergence in certain metrics between
probability distributions for the output of a randomly-initialized (with
gaussian weights) multi-layer perceptron (MLP) to its gaussian process limit.
They are specifically interested in using second-order Poincar\'{e}
inequalities to establish these rates (in the 1-Wasserstein,
Kolmogorov-Smirnov, and total variation distances), a tool developed in prior
works by Chatterjee 2009 and Vidotto 2020 which reduce these estimates to the
calculation of various norms of gradients and Hessians. The authors results
thus apply to networks with $C^2$ activations. The authors show rates for
shallow networks that reflect the "right" dependence on the width of the
network; they also present rates for deeper networks, which do not have the
"right" rate ($n^{-1/4}$ instead of $n^{-1/2}$). Some numerical results are
presented which illustrate the results for shallow MLPs.

**Audience:**

Yes

**Broader Impact Concerns:**

None.

**Claims And Evidence:**

Yes

**Requested Changes:**

- Additional motivation (both for the general problem of studying random neural
  networks, and for the use of second-order Poincare inequalities in
  probability) in order to better contextualize the importance of the authors'
  contributions.
- A more precise discussion of the negative results in section 4, again to make
  the TMLR publication of the authors' negative results more useful for future
  work.

**Strengths And Weaknesses:**

- The contribution of the work is somewhat narrow, given that for the setting
  of GP behavior of MLPs (i.e., no dynamics), the case of deep networks is of
  most interest -- the results that apply to the deep setting mainly assert
  that the main technical tool being applied, second-order Poincar\'{e}
  inequalities, do not give the correct rates here. Hence it seems like the
  main contribution is to demonstrate that these tools are not viable for this
  problem; it seems like this contribution would be showcased as more impactful
  if the authors spent a bit more on motivation (why are these tools important)
  in the intro (some bullets about this below). It would also be helpful if the
  discussion of the multi-layer case in section 4 was less heuristic: for
  example, it would be helpful if it was stated concretely whether the failure
  to get the correct rates in the $L=2$ setting is due to some estimates used
  by the authors that are loose, or whether the authors are simply computing
  the RHS of Vidotto's bound and this bound is not good enough. It seems like
  this latter situation would be of general interest, but it does not seem to
  me that the current submission does enough to highlight "what is going wrong"
  here and what might be necessary to achieve the right rate -- in this
  connection, it seems like it would be helpful if some specific examples were
  discussed (e.g., does Vidotto's bound already fail for polynomial
  activations, where everything can be computed in closed form? and is it
  possible in this setting to directly calculate what the rate for TV/KS/W1
  should be?)


- The introduction would be improved by including additional motivation or
  background at the start -- the current presentation, which begins with the
  notation (some of which will not even be necessary to read the main results
  in the rest of the section, esp. 1.1), feels somewhat unnecessarily
  inaccessible. Similarly, it might make sense to begin by only presenting the
  minimum notation necessary to state the main contributions, before including
  the detailed overview of notation and precise definitions later on.

- I feel that some improved technical motivation in the introduction could help
  contextualize the contributions of the work as well -- for example, some
  discussion of previous uses of second-order Poincar\'{e} inequalities beyond
  the background given in section 2 that suggests why it is important to
  investigate these in the present context (have they been seen to give sharp
  rates in other problems?  etc.)

- The work would benefit from a careful proofreading for typos (see a selection
  of typos below).


### Questions/Clarifications

- Equation (2): Forgive me if I have missed something, but I was not able to
  find this level of resolution in this expression in the cited work -- in
  particular, the dependence on the parameter $p$. The cited work's Theorem 3.1
  does not seem to explicitly denote the rate with respect to the dataset
  $\mathcal{X}$ and its dimensionality (etc.); it is also hard for me to see
  how the expression for the constant given in the cited work's equation (3.7)
  allows to read this off (if it can be done with a bit of effort, and this is
  where your Equation (2) comes from, I would appreciate some clarifying
  details).
- The discussion on page 8 about algebraic complexity of gradient and hessian
  calculations for deep networks is mostly heuristic; is it possible to
  supplement this discussion with any concrete comparisons to prior work, eg
  Hanin's work on these recursive formulas?  (Correlation functions in random
  fully connected neural networks at finite width)
- Is it possible to say anything more precise about the scales of the
  maximum/minimum eigenvalues of the covariance matrix that appear in Theorem
  3.3 in some cases of interest (e.g., random data on the sphere or unit ball
  or hypercube)? The references given for estimating eigenvalues seem to apply
  to graph Laplacians or transition matrices for Markov chains; are these also
  relevant to the covariance matrix $C$ in the present context?
- Some related work not discussed which might be of interest (it seems that
  this line of work is not discussed in any of the "quantitative CLT" prior
  works) are those works that develop subgaussian-type tail bounds for various
  quantities associated to random neural networks in parameter regimes of
  interest (e.g., wide networks or deep networks). For example, it seems like
  something like [1], Lemma D.1 (or more generally Lemma D.10) could be
  combined with a truncation argument and the fact that the $W_2$ distance
  between gaussians is bounded by the operator norm difference between their
  square-root-covariance matrices to get control in $W_1$ of order $C \sqrt{L
  p^2 / n}$, where $C$ is an absolute constant (and log factors are
  ignored); unfortunately this result only applies to ReLU activations with
  critical $\sigma_{w}$ and no biases, though. These kinds of results may be of
  interest because they seem to specify a superior dependence on the depth $L$
  than other results mentioned; on the other hand, they may be limited by the
  dependence on $p$ (since this type of argument controls the operator norm
  difference between covariance matrices by the Frobenius norm).

[1] https://openreview.net/forum?id=O-6Pm_d_Q-


### Minor issues (typos, etc)

- Notation: I would recommend using a notation for the inputs that is "more
  contrasting" than \mathbf for rows and \boldsymbol for columns, unless this
  is a standard notation that I am not aware of. For example, what about raised
  indices for rows (or vice versa), or capital-with-a-subscript for columns and
  lowercase for rows.
- Related work discussion (quantitative CLTs part): I could not see how the
  cited work by Apollonio et al is a strict "generalization" of the result of
  Basteri and Trevisan (please correct me if I am mistaken, but it seems that
  the results of Apollonio et al. only hold when $p=1$).  In the same sentence,
  writing "...(presumably) better constants", I would recommend to simply
  delete this quoted part if it is not immediately clear whether the constants
  are improved.
- Last paragraph on page 2: there is a spurious capitalization of "Uniformly"
  in the middle of a sentence
- Page 3: in first paragraph, what are "presumably optimal rates"? Preferable
  to state that they are optimal, if so; if not, delete this clause?
  This usage appears again on page 5 in the first paragraph.
- Page 3: last paragraph on page: "leading to a worst rate" -> "worse rate";
  this usage appears again at the top of page 8
- Start of section 2: should there be an absolute value (or a euclidean norm)
  around $X$ in the definition of the $L^q$ norms?
- Theorem 2.3: it might be preferable to maintain a consistent notation for
  this matrix norm -- in (6), it is labeled as the "operator norm", but here
  "spectral norm"?
- Page 11 last paragraph: "...use a whatever $\tau$" ?? (arbitrary?)
- The tone in parts of the last paragraph of the submission (top of page
  12/bottom of page 11) does not seem to be completely appropriate for a TMLR
  submission ("we try to derive a specific bound for the ReLU", "...the idea
  would have been... but... the limit exploded"), although I appreciate wanting
  to include these negative results; possibly this could reworded by simply
  mentioning that it does not work to use these kinds of approximating
  sequences and "pass to the limit" because (as you write later) "...Theorem
  A.2 needs each..." (hence just splicing these two together).

---

### Review · Reviewer_n17D · 2023-09-07

**Summary Of Contributions:**

The paper introduces the use of second order Poincare inequalities to establish bounds on central limit theorem convergence of shallow and deep neural networks to Gaussian processes. The convergence result for shallow networks is classic due to Neal, with convergence for deep networks more recent. There is also prior work on convergence rates for shallow and deep neural network, establishing an $n^{-1/2}$ convergence rate for both, where $n$ is the width of the neural network. The present paper replicates the convergence rate result for shallow networks using the new second order Poincare inequality approach. For deep networks, a weaker $n^{-1/4}$ result is obtained. The authors discuss reasons for the weaker result and also provide simulations to confirm the bounds, using implicit and explicit approaches.

**Audience:**

Yes

**Claims And Evidence:**

Yes

**Requested Changes:**

Most importantly: I think the authors should strongly consider clarifying the contribution. The approach is new and interesting. Why should people care, given that the results are weaker than what is already in the literature. It would seem important to argue that this approach offers a promising new direction for addressing some open problem? Otherwise, it is not clear that there is much value added by the paper to the existing literature.

Please see detailed comments for additional requests related to clarifying contributions and wording.

**Strengths And Weaknesses:**

Strengths:
- The topic, understanding the effects of width on deep neural networks is important and of great interest.
- The technical results are nice and the approach is very interesting.
- The exposition of the main paper is generally clear.
- The authors do a nice job reviewing the literature and describing prior results.

Weaknesses:
- The paper is primarily methodological, given that the main results exist in stronger forms elsewhere.
- Given that the results already exist using different approaches, I would have expected the authors to make a stronger case for why this approach could be a useful contribution to the literature.
- There are a number of more minor matters (see below).

Detailed comments:
- " depth L ≥ 1 and width n ≥ 1, a well-established result is that, as n → +∞, the NN’s output converges in distribution to a Gaussian process" It would be good to cite someone for this well-established result in the abstract.
- "we investigate the use of second-order Gaussian Poincaré inequalities to obtain quantitive CLTs for the NN’s output, showing their pros and cons in such a new field of application" This statement is more vague than would be ideal as a summary of contributions.
- "Such a worsening in the rate is a peculiar feature of the use of second-order Poincaré inequalities, which are designed to be applied directly to the NN’s output as a function of all the previous layers, hence not exploiting the recursive structure of the NN and/or its infinitely
wide Gaussian limit. While this is a negative result over the state-of-the-art, it does not diminish the effectiveness of second-order Poincaré inequalities, which we prove to maintain their effectiveness in establishing a quantitative CLT for a complicated functional of Gaussian processes such as the
deep Gaussian NN." I don't understand the claim here.
- "with (presumably) optimal rates" what does this mean?
- A detailed introduction to second-order Poincare inequalities would be helpful for the reader. Section 1.1 is hard to parse without a clear introduction, especially because this seems to be a method paper focusing on that tool.
- Numerous times throughout the paper, a seemingly important word is included in parentheses, e.g. (algebraic) and (implicit) and (presumably). I do not know how to interpret these statements.
- "While this is a negative result over the state-of-the-art in the field, it does not diminish the effectiveness of second-order Poincaré inequalities, which we prove to maintain their effectiveness in establishing a quantitative CLT for a complicated functional of Gaussian processes such as the deep Gaussian NN." I don't understand this statement.
- "which leads to move further away from the distance" Minor grammar issue.
- "analogous result follow from" grammar.
- "depends on some expectations of the standard Gaussian law" This is rather vague for a theorem.

---

### Decision · Action_Editor_Snx1 · 2023-10-23

**Recommendation:** Reject

**Comment:**

Unfortunately, reviewers agree that accepting the paper is difficult due to a stronger result in the context. The existence of such a result raises questions about the contribution of the work, which were not clearly addressed in the authors' rebuttal. Therefore, the paper's position in the literature and its motivation are unclear. We do not think that the paper needs expansion and improvement of the technical content. However, we expect authors to clarify the relationship with related studies and the advantages and limitations of using Poincaré inequalities. This would require a major paper revision. For example, the reviewers suggest the following revisions:

- The authors should provide a more robust characterization of why it is important to understand the (failure of) applicability of second-order Poincaré inequalities for studying this problem for deep networks. General improvements in writing are also necessary.
- To justify the authors' approach, they should elaborate on the significant connection with [Hanin (2022)], which, a priori, applies to deep networks and achieves better results than the second-order Poincaré approach. In the response, the authors mentioned that [Hanin (2022)] may apply to a different setting than the authors’ analysis. If this is indeed the case,  the authors should carefully explain in the paper how this is so. This would support the usefulness of the authors' analysis for the future.
- While the claims provided in the responses could justify this work, the authors should provide more detail about the applicability of the proposed framework to a broader range of networks and the limitations of the existing approach.

> Gaussian limits are available for other NN’s architectures (see Yang, 2019), and of quantifying these limits is a natural problem. Convolutional NNs are arguably the most popular example, for which a Gaussian limit has been obtained in Novak et al. (2018).

> In general, we believe that extending the approaches of Basteri & Trevisan (2022) and Favaro et al. (2023) to other NN’s architectures would be a non-trivial problem, probably not even feasible. This motivates the importance of our work in the context of quantitative central limit theorem for NNs. We can make clear this point in the paper.

**Audience:**

The main weakness raised by reviewers is the existence of a stronger result. Whereas the use of Poincaré inequality seems technically interesting, the advantage of using this technique is unclear. In order to draw the attention of the readers, it should be further explained why this approach could contribute to the literature.

**Claims And Evidence:**

This work studies the convergence (Central Limit Theorem) of randomly initialized shallow and deep neural networks to its Gaussian process. The authors introduce the use of second-order Poincaré inequalities [Chatterjee (2009) and Vidotto (2020)] to establish the convergence rates in several metrics. Specifically, they derive the rate of $n^{-1/2}$ for shallow networks and derive a slower rate of $n^{-1/4}$ for deep networks. These convergence rates are empirically verified by some experiments.

**Resubmission Of Major Revision:**

The authors may consider submitting a major revision at a later time.